# The effect of military training on the sense of agency and outcome processing

Emilie A. Caspar [1,2✉], Salvatore Lo Bue [3], Pedro A. Magalhães De Saldanha da Gama[1], Patrick Haggard [4,5] & Axel Cleeremans [1,5]

Armed forces often rely on strict hierarchical organization, where people are required to follow orders. In two cross-sectional studies, we investigate whether or not working in a military context influences the sense of agency and outcome processing, and how different durations (junior cadets vs senior cadets) and types (cadets vs privates) of military experience may modulate these effects. Participants could administer painful electrical shocks to a 'victim' in exchange for money, either by their own free choice, or following orders of the experimenter. Results indicate that working in a strictly hierarchical structure may have a generalized negative impact on one's own sense of agency and outcome processing by reducing it, even when participants could freely decide their action. However, trained officers showed an enhanced sense of agency and outcome processing. This study offers insights on the potential for training the sense of agency and outcome processing.

[1] Consciousness, Cognition and Computation Group (CO3), Center for Research in Cognition & Neurosciences (CRCN), ULB Neuroscience Institute (UNI), Université libre de Bruxelles (ULB), Brussels, Belgium. [2] Social Brain Lab, The Netherlands Institute for Neurosciences (NIN), KNAW, Brussels, Belgium. [3] Department of Behavioral Sciences, Royal Military Academy (RMA), Brussels, Belgium. [4] Institute of Cognitive Neuroscience, University College London (UCL), London, United Kingdom. [5] These authors contributed equally: Patrick Haggard, Axel Cleeremans. ✉email: ecaspar@ulb.ac.be

Some social structures, such as the armed forces, rely on a strict hierarchical organization where people are required to follow orders. Historical precedents[1] and early experimental studies[2,3] drew attention to how markedly coercive orders can transform normal patterns of behavior. Milgram[2] reported that 65% of his volunteer participants administered what were described to be dangerous levels of electrical shock to another individual they had just met few minutes before, simply because they had been ordered by an experimenter to do so for the sake of the experiment.

In normal circumstances, society assumes that individuals are responsible for the outcomes of their actions. However, individuals are also expected to conform to more-or-less rigid social rules governing behavior[4]. Individuals are thus constantly in balance between taking full responsibility for their own actions, and subjugating their personal action choices to social forces, at which point the notion of personal responsibility seems less pertinent[5]. Military personnel acting under command represent an extreme of this balancing act, since their professional role implies compliance to hierarchical authority, based on the mandate society has given to that authority. The structure of most armies is based on strict hierarchies where people have a professional legal duty to follow orders. At the same time, they also have a professional legal duty to disobey illegal orders[6]. In recognition of this unusual structure of responsibility, military personnel acting under orders are often considered differently from other agents in discussions of responsibility. In particular, under some circumstances, though certainly not all, "only obeying orders" may be accepted as a basis of diminished responsibility. Here we ask how the highly hierarchical and even coercive structure of the military affects the experience of autonomy and agency.

Being an autonomous agent includes the subjective experience that one is the author one's own actions and their consequences[7], and is thus responsible for what one chooses to do. The experience of agency is therefore central to responsibility and autonomy, yet it is hard to study scientifically. Explicit reports of "sense of agency" are problematic, however, because of strong self-efficacy and self-serving biases[8]. We, therefore, used an alternative, implicit measure of sense of agency, based on time estimation of action-outcome intervals (refs. [9,10] for reviews). The relationship between time perception and sense of agency is mediated by the involvement of striatal dopaminergic activity, which is crucial for time perception, e.g., refs. [11,12], and which also contributes to the basal ganglia drive to frontal motor areas, e.g., refs. [13,14], key brain regions in generating a sense of agency, e.g., refs. [15,16]. In studies asking participants to estimate the duration of intervals between an action and its predictable outcome, participants report shorter interval estimates when the action was performed voluntarily than when this action was performed involuntarily, for instance after a TMS pulse over the motor cortex[17]. Moreover, this "intentional binding" effect is stronger for actions involving choices that are meaningful to the agent, compared to comparable actions without any element of choice[18,19]. Estimates of the action-outcome interval are therefore a valuable, implicit proxy measure of the sense of agency.

In a recent study, we showed that social coercion also affects individuals' experience of agency and responsibility[20]. In this paradigm, two volunteers respectively took on the role of the agent or the "victim". In one condition, agents were free to choose between administering or not a painful electric shock to the "victim" in order to increase their own remuneration. In another condition, the experimenter ordered the agent to administer or not the shock. In order to evaluate agents' sense of agency, each key press triggered a tone and participants had to estimate the delay between that key press and the resulting tone (i.e.,

"intentional binding" effect). Overall, results showed that being ordered to perform an action reduced the experience of being the intentional agent of the action, in comparison with being free to choose that action. In addition, electrophysiology results showed that the amplitude of the auditory N1, an evoked-related potential associated with auditory stimuli (i.e., the tone triggered after each key press), was reduced in the coercion condition. This suggested that coercive instructions reduce the sensory processing for action outcomes (i.e., outcome processing). This could perhaps explain why obeying orders can influence behavior in social settings, e.g., refs. [21,22]. We coined this phenomenon of reduction of agency and outcome processing under command the "coercion effect". Note that a strict definition of "coercion" would refer to the use of force, or threat of force, to persuade someone to do something that they are unwilling to do—but this cannot and should not be studied experimentally, since it clearly violates ethical codes. Here, we use the conventional term "coercion" to refer to an experimental situation in which people obey orders to inflict painful stimulation to another individual.

Previous work on the "coercion effect" was conducted on civilian volunteers, for whom autonomy might be greater than for military personnel. In a first study, we assessed whether people working in a military context would have a different experience of autonomy under coercion compared to a non-military control group. In this study, participants were either free to decide, or were ordered by an experimenter, to deliver (or not) a painful shock to the "victim". We measured the sense of agency with the method of interval estimates following each action, and explicit responsibility ratings for the entire episode. The sense of moral responsibility is often considered as a relatively high-level conceptual representation of the self-in-action in a given social context[23]. Outcome processing was measured with the amplitude of the auditory N1. Previous studies suggest that sense of agency and outcome processing are related because similar factors influence them, e.g., refs. [24,25] but reliable statistical correlations between those measures have barely been reported[26]. Similarly, implicit and explicit measures of agency represent different levels of representations of the self-in-action[27], with distinctive neural bases[15,28–32]. A military environment could thus have a different effect on outcome processing and on the sense of agency. In the coercion condition, orders were given either by a ranked officer or by a civilian experimenter, in order to investigate to what extent the identity of the experimenter has a role in the coercion effect.

Results indicate that working in a highly hierarchical organization negatively impacts the sense of agency, both under free-choice and coercion. Further, prolonged military experience as a subordinate has a negative impact on both sense of agency and outcome processing. However, trained officers did not show this effect, perhaps reflecting the accountability and personal responsibility associated with their military role.

## Results
**Study 1.** Behavioral results relative to the number of shocks participants freely delivered to the "victim" are reported in Supplementary Notes 1.

*Interval estimates:* We conducted a repeated-measures ANOVA on agents' interval estimates, with Condition (free-choice vs. coercion) as within-subject factor and Group (junior cadets vs. civilian students), Experimenter (ranked experimenter vs. civilian experimenter), and Order of the Role (agent first vs. "victim" first) as between-subject factors. Because in the present study we aimed to compare different groups of volunteers and because it is known that participants may differ in the way they use the scale to provide an answer (estimates between 1 and 1000 ms), we used *z*-score transformed data. *z*-scores reduce irrelevant

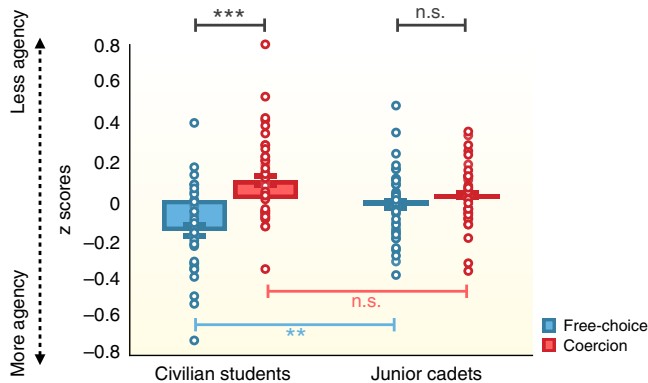

**Fig. 1 z-scored interval estimates, used as an implicit measure of sense of agency, in free-choice (blue) and coercion (red) conditions for both junior cadets (n = 38) and student civilians (n = 39).** Graphical display showing the interaction Condition*Group in a mixed ANOVA with independent (red, blue significance lines) and paired (black significance lines) sample t-tests. All tests were two-tailed. Data are presented as mean values ± SEM. Errors bars represent standard errors. *** represents a p-value ≤ 0.001. ** represents a p-value between 0.001 and 0.01. n.s. indicates a non-significant result. z-scores were higher for the group of junior cadets than for the group of civilian students in the free-choice condition (p = 0.005). Colored dots represent individual participant data. Source data are provided as a Source Data file.

inter-subject variability by subtracting from each interval estimate the mean estimate for that participant across all trials and by dividing the resulting differences by the standard deviation of all estimates for that participant. As with raw interval estimate data, a lower z-score is taken to imply a higher sense of agency. We observed a significant effect of Condition ($F(1,69) = 12.335$, $p = 0.001$, $\eta^2_{partial} = 0.152$), a main effect of Group ($F(1,69) = 7.752$, $p = 0.007$, $\eta^2_{partial} = 0.101$) and a significant interaction between Condition and Group ($F(1,69) = 5.960$, $p = 0.017$, $\eta^2_{partial} = 0.080$) (Fig. 1). Paired-comparisons indicated that interval estimates were significantly lower in the free-choice condition (z-scores: $-0.168$, $CI_{95} = -0.247$ to $-0.089$) compared to the coercion condition (z-scores: $0.106$, $CI_{95} = 0.034–0.177$) for the civilian student group only ($t(38) = -3.791$, $p = 0.001$, Cohen's $d = 0.608$), indicating a coercion effect on the sense of agency. For junior cadets, this difference was not significant ($p > 0.3$). To ensure that the presence of a coercion effect on interval estimates for civilian students and the lack of coercion effect for junior cadets was not simply due to a lack of sensitivity in our data, we further computed Bayes Factors (BFs)[33]. A BF between 1/3 and 3 indicates a lack of sensitivity. A BF below 1/3 or above 3 is typically interpreted as support for the null hypothesis, or for the alternative hypothesis, respectively. For civilian students, the $BF_{10}$ was 54.98, thus supporting the difference between the free-choice and the coercion condition. For junior cadets, the $BF_{10}$ was 0.257, thus supporting H0.

We then evaluated whether this lack of coercion effect in the cadet group was due to high interval estimates in the free-choice condition (which would imply a reduced sense of agency in normal autonomous action) or due to low interval estimates in the coercion condition (which would imply a preserved, high sense of agency under coercion) in comparison with the group of civilian students. We thus conducted independent sample t-tests between the two groups on interval estimates in both the free-choice and the coercion conditions. We observed that interval estimates were significantly shorter for civilians (z-scores: $-0.168$, $CI_{95} = -0.247$ to $-0.089$) than for junior cadets (z-scores: $-0.026$, $CI_{95} = -0.086–0.033$) in the free-choice condition ($t(75) =$

$-2.893$, $p = 0.005$, Cohen's $d = -0.659$), but that this difference was not significant in the coercion condition ($p > 0.06$). This suggests that junior cadets exhibited a reduced sense of agency in the free-choice condition in comparison with civilian participants. None of the other main effects or their interaction were significant (all $ps > 0.1$). Results of explicit ratings of responsibility are reported in Supplementary Notes 2.

The observed reduced sense of agency in the free-choice condition for junior cadets could be the result of their military experience, or could reflect a predisposing trait favoring joining military organizations. We, therefore, conducted a range of supplemental analyses reported in Supplementary Notes 3. Crucially, while we found some evidence for lower impulsivity and higher social dominance in the military cadets, compared to the control group, we found no evidence that these traits were linked to our main measure of the coercion effect, using full sample regression analyses. In addition, we observed that the degree to which civilians thought that they would be suited to military environments did not influence the results. In short, we did not find convincing evidence that a reduced sense of agency in people who choose to undergo military training is related to intrinsic predispositions or personality traits.

*EEG recordings*: All actions were always followed by an auditory tone, whether a shock occurred or not. Analysis of the auditory N1 evoked by the outcome tone was measured as mean amplitude across Fz, FCz, and Cz[24]. We determined the auditory N1 amplitude as the most negative peak within the 90–170-time window. We conducted a repeated-measures ANOVA, with Condition (free-choice vs. coercion) as within-subject factor and Group (junior cadets vs. civilian students), Experimenter (ranked experimenter vs. civilian experimenter) and Order of the Role (agent first vs. "victim" first) as between-subject factors on the auditory N1. Data of five participants were unusable due to a faulty electrode. The main effect of Condition was significant ($F(1,64) = 24.936$, $p < 0.001$, $\eta^2_{partial} = 0.280$), with coercion leading to smaller auditory N1 amplitudes than free choice ($-7.1$ μv, $CI_{95} = -8.1$ to $-6.1$ and $-9.2$ μv, $CI_{95} = -10.2$ to $-8.2$) (Fig. 2). No other factors significantly influenced the effect of Condition (all $ps > 0.3$). No other main effects or interactions were significant (all $ps > 0.3$).

**Study 2.** Given that the type of experimenter did not influence the results in Study 1, this factor was not included in Study 2. Participants were only tested by a senior military captain, similarly to Study 1. Behavioral results relative to the number of shocks participants freely delivered to the "victim" are reported in Supplementary Notes 4.

*Interval estimates:* We conducted a repeated-measures ANOVA on agents' interval estimates transformed in z-scores, with Condition (free-choice vs. coercion) as within-subject factor and Group (junior cadets vs. seniors vs. privates), and Order of the Role (agent first vs. "victim" first) as between-subject factors. We observed a significant effect of Condition ($F(1,79) = 7.295$, $p = 0.008$, $\eta^2_{partial} = 0.085$) and a significant interaction between Condition and Group ($F(2,79) = 3.256$, $p = 0.044$, $\eta^2_{partial} = 0.076$) (Fig. 3). Paired-comparisons indicated that the difference between the free-choice condition and the coercion condition was significant for the group composed of seniors ($t(29) = -3.271$, $p = 0.003$, Cohen's $d = 0.597$), with lower interval estimates in the free-choice condition (z-scores: $-0.127$, $CI_{95} = -0.207$ to $-0.047$) than in the coercion condition (z-scores: $0.087$, $CI_{95} = 0.024–0.149$), thus implying a coercion effect in that group. This difference was neither significant for the group of junior cadets ($p > 0.4$), thus replicating Study 1, nor for the group of privates ($p = 0.6$). Independent sample t-tests for each condition between

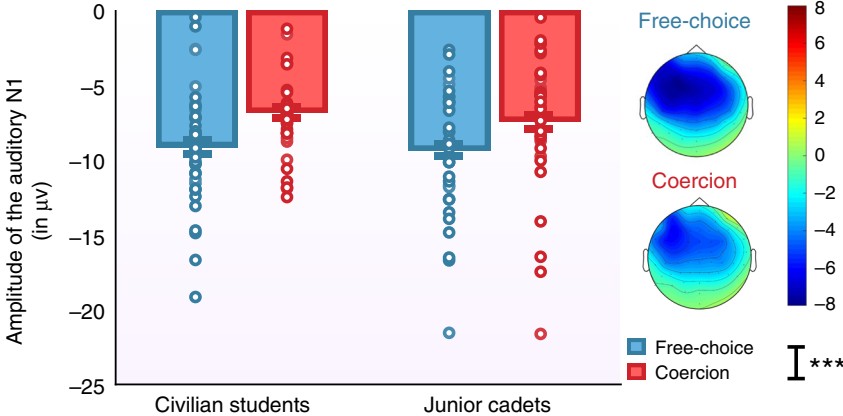

**Fig. 2 Mean amplitude of the auditory N1 in both the free-choice (blue) and the coercion (red) conditions, respective to the Group (junior cadets – $n$ = 36 vs. civilian students – $n$ = 36).** Data are presented as mean values ± SEM. Errors bars represent standard errors. Graphical display showing the interaction Condition*Group in a mixed ANOVA with the main effect of Condition represented by a black significance line. All tests were two-tailed. *** represents a $p$-value < 0.001. Colored dots represent individual participant data. Source data are provided as a Source Data file.

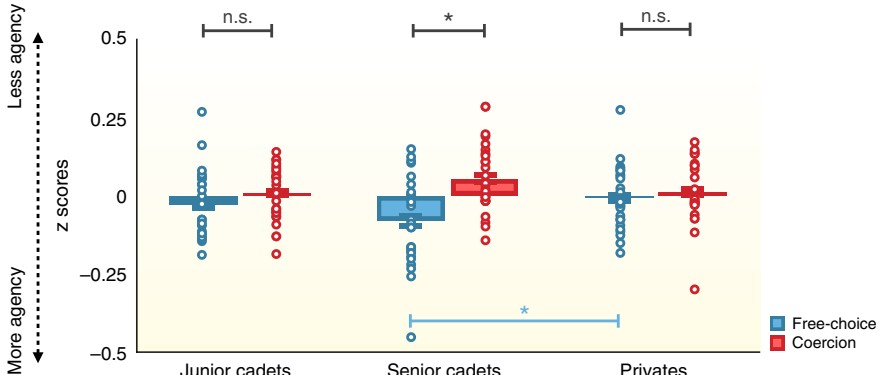

**Fig. 3 $z$-scored interval estimates, used as an implicit measure of sense of agency, in free-choice (blue) and coercion (red) conditions for the three groups: junior cadets ($n$ = 28), seniors ($n$ = 30), and privates ($n$ = 27).** Data are presented as mean values ± SEM. Errors bars represent standard errors. Graphical display showing the interaction Condition*Group in a mixed ANOVA with independent (red, blue significance lines) and paired (black significance lines) sample $t$-tests. All tests were two-tailed. *** represents a $p$-value ≤ 0.001. * represents a $p$-value between 0.01 and 0.05. n.s. indicates a non-significant result. $z$-scores were lower in the free-choice condition for senior cadets than for privates ($p$ = 0.025). Colored dots represent individual participant data. Source data are provided as a Source Data file.

groups revealed that interval estimates were significantly lower in the free-choice condition for seniors ($z$-scores: −0.127, $CI_{95}$ = −0.207 to −0.047) than for privates ($z$-scores: −0.01, $CI_{95}$ = −0.073–0.051, $t(55)$ = −2.309, $p$ = 0.025, Cohen's $d$ = −0.613) but not for junior cadets ($p$ = 0.078). Interval estimates for the coercion condition did not differ between any of the groups (all $ps$ > 0.06). Neither the main effect of Group ($p$ > 0.06) nor the other interactions were significant (all $ps$ > 0.1). Results of explicit ratings of responsibility are reported in Supplementary Notes 5.

*EEG recordings:* Analysis of the auditory N1 was measured similarly to Study 1. We conducted a repeated-measures ANOVA, with Condition (free-choice vs. coercion) as within-subject factor and Group (junior cadets vs. seniors vs. privates), and Order of the Role (agent first vs. "victim" first) as between-subject factors on the amplitude of the auditory N1. The data of two participants were not analyzed because of a technical problem with the EEG device during the testing due to a faulty electrode. We observed a significant main effect of Condition ($F(1,78)$ = 9.086, $p$ = 0.003, $\eta^2_{partial}$ = 0.104), a significant main effect of Group ($F(1,78)$ = 4.512, $p$ = 0.014, $\eta^2_{partial}$ = 0.104) and a marginal interaction Condition × Group ($F(2,78)$ = 2.939, $p$ < 0.06, $\eta^2_{partial}$ = 0.070) (Fig. 4). Paired-comparisons indicated that similarly to Study 1, junior cadets had a higher amplitude of the

auditory N1 in the free-choice condition (−10.3 μv, $CI_{95}$ = −11.85 to −8.7) than in the coercion condition (−7.7 μv, $CI_{95}$ = −9.3 to −6, $t(26)$ = −3.900, $p$ = 0.001, Cohen's $d$ = 0.654). The group of seniors ($p$ < 0.7) and the group of privates ($p$ < 0.1) did not display a significant coercion effect. Independent sample $t$-tests revealed that the group of seniors had a higher amplitude of the auditory N1 in the free-choice condition (−10.87 μv, $CI_{95}$ = −12.44 to −9.31) than the group of privates (−7.8 μv, $CI_{95}$ = −9.9 to −5.7, $t(55)$ = −2.423, $p$ = 0.019, Cohen's $d$ = −0.643), but did not differ from the group of junior cadets ($p$ > 0.4). We did not compare junior cadets and privates since they differ both in rank and number of years spent in the military system. In the coercion condition, the amplitude of the auditory N1 was higher for seniors (−10.65 μv, $CI_{95}$ = −12.5−−8.8) than for privates (−6.9 μv, $CI_{95}$ = −8.8 to −5, $t(55)$ = −2.910, $p$ = 0.005, Cohen's $d$ = −0.772) and, than for junior cadets (−7.7 μv, $CI_{95}$ = −9.4 to −6, $t(55)$ = −2.408, $p$ = 0.019, Cohen's $d$ = 0.639). No other main effect or interactions were significant (all $ps$ > 0.5).

Our results in senior cadets might reflect the results of an officer-type training. Alternatively, they could reflect a selection process, in which only junior cadets with a higher sense of agency and stronger outcome processing persevered in the military system. We thus conducted additional analyses. We separated the junior cadets

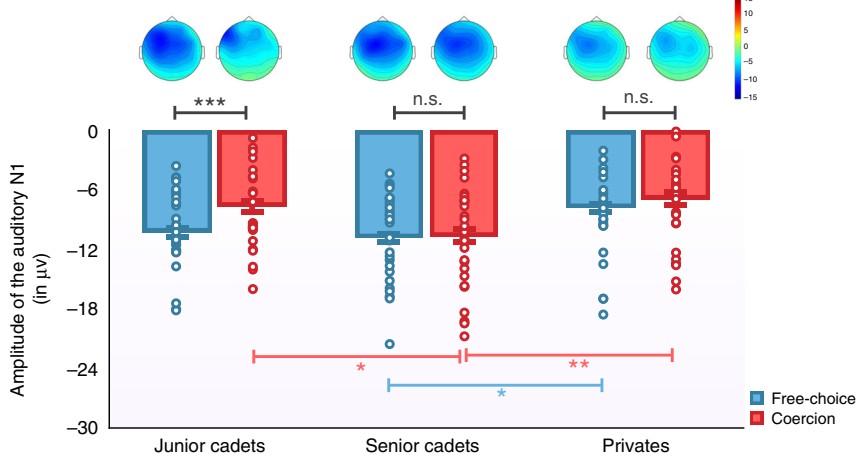

**Fig. 4 Mean amplitude of the auditory N1 in both the free-choice (blue) and the coercion (red) conditions, respective to the Group (junior cadets – $n =$ 28 vs. seniors – $n = 30$ vs. privates – $n = 27$).** Data are presented as mean values ± SEM. Errors bars represent standard errors. Graphical display showing the interaction Condition*Group in a mixed ANOVA with independent (red, blue significance lines) and paired (black significance lines) sample $t$-tests. All tests were two-tailed. ** represents a $p$-value between 0.001 and 0.01. * represents a $p$-value between 0.01 and 0.05. n.s. indicates a non-significant result. The group of junior cadets had a significant difference between conditions ($p = 0.001$). In the free-choice condition, the amplitude of the N1 was higher for senior cadets in comparison with privates ($p = 0.019$). In the coercion condition, the amplitude of the auditory N1 was higher for senior cadets than for junior cadets and privates ($p = 0.019$ and $p = 0.005$, respectively). Colored dots represent individual data. Source data are provided as a Source Data file.

tested in both Studies 1 and 2 into a group who left military training (i.e., group OUT) and the remaining group who persevered in the military system (i.e., group IN). An account of our effects based on selection would predict that cadets who remain within the military system would display a high sense of agency in free-choice, and a high amplitude of the auditory N1 in both experimental conditions, thus making their profile similar to our findings with senior cadets in Study 2. In fact, we observed that junior cadets who remained in the military system displayed a lower sense of agency and a reduction of outcome processing relative to those who left, contrary to the hypothesis of a selection bias. The full results of these analyses are given in Supplementary Notes 6.

## Discussion
In Study 1, we investigated the experience of agency in junior cadets and in a civilian control group. Similar to previous studies[20,34,35], results showed a clear coercion effect for civilian participants. Interestingly, interval estimates of junior cadets were not reliably different in the coercion condition and in the free-choice condition. Importantly, for both groups, these results were not influenced by the identity of the experimenter, suggesting that coercion effects do not depend on the particular social status of the person giving orders, but rather reflect a more general difference between coercion and autonomy contexts[34].

Further analyses revealed that the lack of coercion effect for junior cadets was due to long interval estimates in the free-choice condition, suggesting a reduced sense of agency[9,10]. This implies that junior cadets may show a global reduction of sense of agency, relative to non-military personnel, both when they can freely choose which action to perform and when they are coercively instructed. We also observed that the coercion effect did not appear to be influenced by any personality traits and that $z$-scores in both the free-choice and the coercion conditions were not influenced by the degree to which civilians thought that they would be suited to the military environment. It thus suggests a negative influence of the military environment on cadets' sense of agency, rather than a predisposing trait.

We additionally observed a coercion effect on the amplitude of the auditory N1 suggesting that receiving coercive instructions reduces the neural processing of the outcomes of one's own actions[20]. Importantly, this was the case for both junior cadets and civilian students, suggesting that outcome processing was not impacted by the military environment.

However, Study 1 does not represent a reliable sample of all the different categories of individuals working in the army, nor can it represent the effects of prolonged military training. In Study 2, we, therefore, compared three groups of military personnel, namely privates, junior cadets, and senior cadets. Comparisons between these groups could reveal how prolonged experience in the social environment of a military organization influences the sense of agency under coercion and how the different notions of responsibility enshrined in the training of officers and of ordinary soldiers might lead to modulations of the coercion effect. In comparison with junior cadets, seniors have been trained to be officers during 5 years in average and have reached the rank of lieutenant. They have thus worked for a longer time period within the military than junior cadets, and have been trained to be accountable for their own actions (including giving orders to others) during those 5 years. Privates correspond to troop soldiers. They have a lower rank within the military system and in comparison to cadets, accountability is less emphasized during their training and career, although they work for a similar number of years in this type of organization. One might predict that seniors regain a strong sense of agency when they are free to choose which action to perform given that they frequently command others and are accountable for those actions, e.g., refs. [36–38]. Privates might, on the other hand, continue to exhibit this reduced agency, since they routinely obey orders. One might also predict that outcome processing could be influenced by military rank (i.e., officers vs. privates), as a result of 5 years of differentiated military training including differentiated responsibility. Since the function of privates is mostly to execute orders from the military hierarchy, downregulation of outcome processing could be observed. On the other hand, senior officers might show an upregulation of outcome processing, consistently with

their function implying commanding subordinates and being accountable for their actions.

As in Study 1, in Study 2 junior cadets exhibited a low sense of agency in the free-choice condition. A similar result was also found for the group composed of privates, suggesting that the sense of agency is not positively influenced by the number of years spent in a military organization. Interestingly, we observed that seniors, despite working since a similar number of years in a military organization than privates, appear to have a higher sense of agency in the free-choice condition, similar to civilians. This result offers interesting insights on the possibility to train the sense of accountability, by restoring experience of agency. Indeed, while both junior cadets and privates exhibited a reduction in the sense of agency when they could freely decide which action to perform, hence underlying the negative effect of military hierarchy per se, being trained as an officer appears to block this effect and even reverse it.

We observed that junior cadets displayed a "coercion effect" on the amplitude of the auditory N1, similarly to Study 1. However, this was not the case for seniors and privates. Seniors displayed a high amplitude of the auditory N1 in both the free-choice and the coercion conditions, while privates displayed a low amplitude of the auditory N1 in the same conditions. It thus appears that 5 years in the military decreases the difference in cognitive outcome processing between receiving orders and being free to decide for yourself. Training as an officer appeared to protect against reduction of outcome processing in coercive contexts, while working as a private appeared to have a reduced outcome processing even for freely chosen decision.

Our experiment was designed to identify changes in N1 amplitude due to coercion, and the way that different groups respond to coercion. However, the amplitude of the auditory N1 can be modulated by other factors, such as age, education, and intelligence[39]. For instance, a higher amplitude of the auditory N1 was observed for a high number of years of education. Seniors and privates differ in educational levels: seniors have a university master degree while privates require only an elementary school certificate, thus potentially explaining differences in outcome processing between seniors and privates. However, the amplitude of the auditory N1 in the free-choice condition did not differ between junior cadets and seniors, despite 5 more years of military training and education. It thus seems unlikely that differences between other groups merely reflect differences in the duration of education prior to military service. Other studies showed that the amplitude of the auditory N1 could be modulated by perceptual, motor, and cognitive factors[40]. However, it is also unlikely that those factors influence our results. All participants were in the same age range[41], without any physical disabilities, and heard a tone similar in both frequency and loudness[42]. Also, participants all used the same keyboard and the same fingers to press the buttons, ruling out the influence of motor performance. Attention has been previously discussed as modifying both late and early (i.e., the auditory N1) ERPs. However, it is unlikely that a difference in attention explains the difference between groups since the majority performed correctly the task of interval estimates and did not commit mistakes when pressing the buttons in the coercion condition. Therefore, it seems unlikely that the N1 results in our experimental design are confounded by these other factors, although it cannot be entirely excluded.

We did not investigate personality traits in Study 2. However, previous studies[20,34,35], and supplementary analysis of the present Study 1 did not find strong evidence that personality traits could influence interval estimates and outcome processing.

In the present paper, we investigated the experience of agency in different groups of military personnel and in civilians in order to evaluate the respective influence of working in a military context and being trained to be accountable. We used an implicit measure, based on perceptual compression of action-outcome intervals, as a behavioral marker of sense of agency, and we used a reciprocal, transparent experimental design, in which participants administered electric shocks to another member of their dyad.

The fact that junior cadets and privates show a reduced sense of agency, even when they are free to choose when and what action to perform, sheds light on an important feature of human sense of agency. Being trained to follow orders appears to exert a negative impact on how one experiences agency for one's own actions. It results in an adaptive behavior that reduces the distinction between "receiving orders" into "deciding for oneself". Only if one experiences one's action as voluntary, will one develop a sense of agency with respect to the action's outcome[43]. Therefore, regularly receiving coercive instructions may create a new "normality", in which the experience of acting voluntarily can come to resemble the experience of following orders.

Interestingly, although our data is cross-sectional, it raises the possibility that this effect can be reversed: we observed for the group of senior cadets that being trained to be accountable for one's own action, and the actions of troops under one's commands, was associated with an increased sense of agency, relative to privates and junior cadets who remained in the military system, at least when the individual is free to decide which action to execute. This result offers important insight on the potential for training the sense of agency, in order to avoid potential detrimental effects of a lack of agency. Not considering oneself as the author of an action could lead to moral disengagement, with negative effects on behavioral control[44]. Since the present study is cross-sectional rather than longitudinal, these suggestions can only be tentative, rather than conclusive. Future studies could confirm the role of training emphasizing responsibility on sense of agency and outcome processing by offering a longitudinal approach.

Although obedience is an essential aspect of the efficiency of armed forces, international law state clearly that military members must refuse orders if they do not fit in the interest of the service or if they imply committing a crime[45]. On the one hand, soldiers may have to disengage their moral control to follow orders and take distance from their personal responsibility[46] because the organizational objectives may imply actions that society rejects in peacetime (notably, killing and wounding others). On the other hand, past and recent events show that refusing an illegitimate or illegal order is far from straightforward, and requires considerable personal courage[1,47]. In the present study, we could not investigate to what extent a strong sense of agency helps individuals to resist illegitimate orders, since disobedience rates were extremely low. This question would nonetheless be worth investigating to understand how to manage the risks associated with blind obedience.

Previous studies reported that the sense of agency decreased as the number of alternative possible actions decreased[19,48]. This could explain the reduced sense of agency in the coercion condition, where participants have only one action available, relative to our free-choice condition, where two actions are available. In the present experiment, the response set size was similar for all the groups tested, but only civilians and senior cadets displayed a coercion effect. This suggests that sense of agency under coercion is a matter of how context influences choice, rather than the strict availability of a number of choices. One might also suggest that in our study we did not control for causality between experimental conditions since participants received an auditory instruction in the coercion condition but not in the free-choice condition, which corresponds to real-life situations. However, previous studies involving a verbal instruction in both conditions yet observed the

coercion effect on sense of agency, e.g., ref. [34]. Further, a difference in causality or instruction could not explain differences between groups since all groups were instructed in the same way.

In this study, we did not evaluate the sense of agency for individual actions with an explicit measurement. Explicit trial-wise judgments of agency have generally been used when unexpected action outcomes[49] or uncertainty about who caused the outcome[50] is present. In our study, there is no doubt regarding the outcome of each action, nor is there any ambiguity regarding who causes the outcome of each action. For those reasons, we could not use explicit judgments of agency as our main outcome variable. To explore more reliably whether or not explicit sense of agency is also modulated by the military environment and by the military rank, other experimental designs should be developed.

We also observed that outcome processing, as measured by the amplitude of the auditory N1, was influenced by military rank. Supplementary analyses further revealed that those who persevered in the military system showed stronger reductions in outcome processing, possibility reflecting a capacity to adapt to one's own environment. We suggest that working as a subordinate, as privates and junior cadets must do, may be associated with low levels of outcome processing. In contrast, our findings with senior cadets suggest that being trained to be accountable has a positive effect on outcome processing, since the amplitude of the auditory N1 was high in both conditions for seniors. Officer training may require upregulation of outcome processing, possibly reflecting accountability for one's own decisions and actions, and for decisions and actions of those under one's command.

Taken together, our results also highlight a dissociation between the implicit sense of agency and outcome processing. The psychological, conceptual and biological mechanisms linking both explicit and implicit measures of the sense of agency to neural outcome processing still need to be clarified and addressed in future studies.

Our results suggest systematic differences between groups in sense of agency and outcome processing, as a function of the daily working context but also specific training targeting responsibility. Junior cadets need to first downregulate outcome processing and sense of agency to persevere in the military system. They then start to upregulate those processes as they progress toward become officers. Those who failed in downregulating agency and outcome processing were less likely to remain within the military system after their first year. A capacity to adapt and modulate one's own sense of agency appears to be a key factor for success in working within the hierarchical structures of military organizations. Our results also suggest that military training as an officer reverses this downregulation, perhaps through emphasizing accountability. As a result, officers may possess a sense of agency and processing of outcomes that distinguishes them from those they command. This possibility opens up socially relevant and optimistic perspectives for the development of a culture of responsibility within organizations. Cultures of responsibility within the workplace can enhance both civil society and the military organizations tasked to protect it.

## Methods

**Study 1.** *Participants:* Eighty naïve male participants were recruited in dyads. Forty of these participants were undergoing officer training at the Royal Military Academy of Belgium (RMA), while 40 other participants were following standard university education, mainly at the Université libre de Bruxelles. To estimate the sample size, we used the effect size of Experiment 2 in ref. [20]. In that study, the effect size ($d_z$) was 0.630 (based on the means and SDs of the within-subjects free-choice (mean: 367, SD: 119) and coercion (mean: 426, SD: 131) conditions). To achieve a power of 0.80 for this effect size, the estimated sample size was 22[51]. No previous studies tested the interaction of interest of the present study. We thus set the total sample size to 80 based on the fact that we had two main between-subject factors with two levels each (i.e., the group and type of experimenter—see below). During the recruitment procedure, we ensured that participants were neither close

friends nor relatives, by mixing people studying different academic courses to create the dyads. Participants following a military education were recruited during their first year at RMA (from this point on, we will call this group "junior cadets"). The experiment was timed at the very beginning of the training program, so as to limit the possibility that the two co-participants knew each other too well (since they just started their training). Participants received between €20 and €26 for their participation. The following exclusion criteria were determined prior to further analysis: failure to produce temporal intervals co-varying monotonically with actual action-tone interval, or failure to follow instructions. To identify participants for whom the action-tone intervals did not gradually increase with action-tone intervals, we performed a linear trend analysis with contrast coefficients −1, 0, 1 for the three delays we used. One participant was excluded due to non-significant linear trend analysis. Two participants withdrew their participation during the experiment (when they switched to the role of agent: see Supplementary Notes 7 for qualitative observations). For the remaining 77 participants (39 in the group of civilian students and 38 in the group of junior cadets), the mean age was 20.3 years old (SD = 2.77). Independent sample *t*-testing revealed that the cadets were younger than the civilian students (18.6 years, SD = 0.97 vs. 21.9 years, SD = 2.9; $t(75)=6.759$, $p < 0.001$). All participants provided written informed consent prior to the experiment. The study was approved by the local ethical committee of the Université libre de Bruxelles (permission 008/2016).

*Materials and procedure:* Between 3 and 7 days prior to the experiment, participants completed three questionnaires online: The Interpersonal Reactivity Index[52], the Levenson Self-Report Psychopathy scale[53] and the Social Dominance Orientation scale (SDO)[54]. The order of the questionnaires was counterbalanced across participants.

Two male experimenters conducted the experiment. One was a ranked officer (a senior captain, i.e., a Belgian military rank between those of captain and major) from the Royal Military Academy (RMA) and wore his military outfit during the experimental session. The other experimenter was a senior scientist and wore civilian clothes. Another experimenter was always present in each experimental session to manage the electroencephalogram recording and to determine the pain threshold. Participants were assigned randomly to one or the other experimenter, and the experimenter factor was crossed with the group factor, so that half of the civilian students were tested by the officer, while half of the junior cadets were tested by the civilian experimenter. Civilian participants were tested at the Université libre de Bruxelles (ULB) and junior cadets were tested at RMA. Each participant was thus tested in his own educational environment.

On arrival at the experimental laboratory, participants read an information sheet about the experimental procedure and the aim of the experiment. After reading the document, the experimenter repeated the explanations orally following a standardized procedure, and participants were invited to ask any questions. Afterward, the two co-participants signed their individual consent forms simultaneously, ensuring that they were each aware of the other's consent.

The roles of the participants were assigned randomly, based on where participants happened to sit when they first arrived in the room. One participant started by being the agent and the other participant the "victim". These roles were reversed midway through the experiment, making the procedure fully reciprocal, similarly to the method used by Caspar et al.[20]. The agent and "victim" were seated at a table, facing each other (Fig. 5). A keyboard was placed between them, oriented toward the agent but visible by both. The experimental task ran on a computer located on the agent's right side, with the screen visible only to the agent and to the experimenter. The agent was instructed to press a key on the keyboard at a time he chose after the start of the trial. The keyboard included two keys explicitly labeled: "SHOCK" and "NO SHOCK". Pressing the first one delivered a painful electric shock to the victim; pressing the second one delivered no shock.

Shocks were delivered using a constant current stimulator (Digitimer DS7A) connected to two electrodes placed on the back of victims' left hand, visible to the agent. Participants' individual pain threshold was determined for the two participants after they had signed the consent form, before starting the experiment. This threshold was determined by increasing stimulation in steps of 1 mA[20]. We approximated an appropriate threshold by asking a series of questions about their pain perception during the calibration (1. « Is it uncomfortable? » - 2. « Is it painful? » - 3. « Could you cope with a maximum of 100 of these shocks? » - 4. « Could we increase the threshold? » - 5. « On a scale from 0 to 10, where 0 is not painful at all and 10 is the worst possible pain you can imagine; how would you rate this stimulus? »). When roles were reversed, we briefly re-calibrated the pain threshold of the new victim by increasing the stimulation again from 0 in steps of 3 mA up to the previously determined threshold, to confirm that the initial estimate was still appropriate, and to allow re-familiarization. The mean stimulation level selected by this procedure was 34.7 mA (SD = 16; for a pulse duration of 200 μs). This procedure ensured that both participants knew how painful the shocks were and were fully aware that shocks were real.

Whether the agent delivered a shock or not, a 400 Hz tone occurred after the key press. The loudness, duration and intensity of the tone was similar for all participants in each experimental condition. The delay between key press and tone was set to vary randomly at 200, 500, and 800 ms. If a shock was delivered, it occurred at the same time as the tone. A visible muscular twitch on the victim's hand was observable by the agent when the shock occurred. The participants' task was to estimate the delay between the agent's key press and the tone (i.e., intentional binding[55,56]). Participants were informed that the delay would vary randomly on a

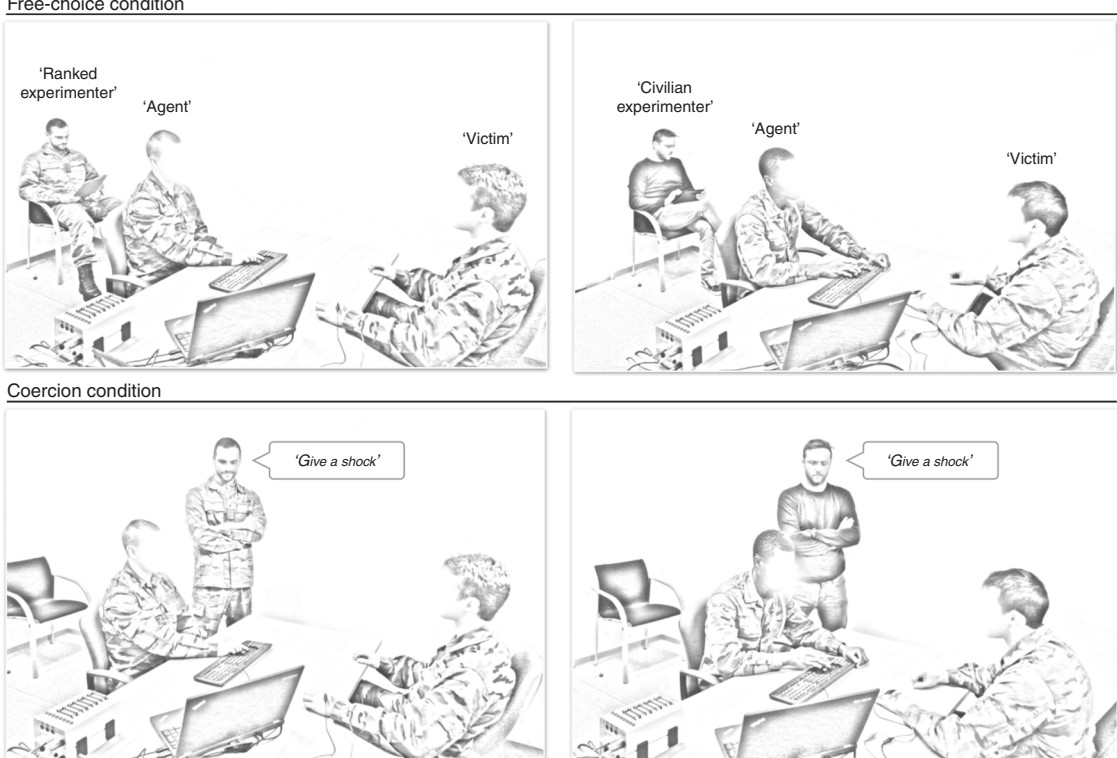

**Fig. 5 Schematic representation of the experimental set-up and procedure for the group composed of junior cadets.** Note the use of a military experimenter for some dyads, and a civilian experimenter for others. The set-up and the procedure were similar for the group composed of civilian students.

trial-by-trial basis, between 1 and 1000 ms (they were reminded that 1000 ms equals 1 s). Participants were also told (1) to make use of all possible numbers between 1 and 1000, as appropriate, (2) to avoid restricting their answer interval, and (3) to avoid rounding. Each participant received a paper sheet with 60 empty boxes in which to write their time estimates in each condition of the task. Participants' answers were hidden from view of the other participants by a cardboard divider, so as to avoid participants being biased by the other participant's answers.

Brain activity was recorded using a 64-channels electrode cap with the ActiveTwo system (BioSemi) and data were analyzed using Fieldtrip software[57]. The activities from left and right mastoids and from horizontal and vertical eye movements were also recorded. Amplified voltages were sampled at 2048 Hz. Data were referenced to the average signal of the mastoids and filtered (low-pass at 50 Hz and high-pass at 0.01 Hz). Artefacts due to eye movements were removed based on a visual inspection with the removal of epochs containing eye blinks or ocular saccades. Because of the EEG recordings, agents were further instructed to wait a minimum of 2 s in a relaxed position before pressing a key, so as to obtain a consistent and noise-free baseline taken $-500$ to $-300$ ms before the occurrence of the tone. Participants were additionally instructed not to move for up to 1 s after the tone and asked to avoid blinking when they pressed a key.

A short initial training session allowed participants to practice the interval estimate procedure without any shocks. All participants started with a specific amount of money, i.e., €20, for their participation. In the free-choice condition, agents were instructed that they could freely choose to increase their remuneration for the experiment by delivering a painful electric shock to the "victim", using the appropriate key on the keyboard. They were told that they were totally free to choose how to act. The agents earned €0.05 each time they decided to deliver a painful electric shock to the "victim". They earned no extra money if they decided to press the "NO SHOCK" key. In the coercion condition, the experimenter stood next to the agent and ordered him, on each of the 60 trials, whether he had to deliver a shock or not, in other words, to press one or the other key (Fig. 5). To have a comparable outcome effect (shock vs. no shock) in the analysis, the agents again received €0.05 each time they administered a shock in the coercion condition. The experimenter ordered the agent to deliver a shock on 30/60 trials, in a haphazard order, based on his own decision. There were 60 trials per condition (20 trials for each action-tone delay, in randomized order), resulting in a total of 240 trials (120 per role). The order of free-choice and coercion conditions was counterbalanced but similar within each dyad, meaning that the order of conditions was the same for the two participants in a dyad. The Psychtoolbox on MATLAB was used to display the experiment and SPSS was used to analyze the data.

In a post-session questionnaire, participants were invited to estimate in percent how much responsible they felt in each experimental condition. They were also invited to describe in a few words what they had felt during the experiment, and any reactions they had to the experiment. Finally, participants were paid separately based on the financial gain earned during the experiment. Also, we re-contacted the civilian volunteers after the experimental session, to ask them a series of three additional questions: "To what extent do you think that you would be suited for military training?" Answers from 0 (not at all) to 10 (entirely)—"Have you ever considered joining the army?" (Answer from 0 (never) to 10 (frequently)) —"Would it be difficult for you to work in a highly hierarchical environment?" Answer from 0 (very difficult) to 10 (not difficult at all). These questions were designed to investigate whether prior interest in working in a military system, or putative underlying trait factors that might predispose to joining military organizations, would influence the results.

**Study 2**. *Participants*: Ninety new naïve male participants were recruited in dyads. Thirty were junior cadets, thirty were seniors and thirty were privates. Based on the similar exclusion criteria than those we used in Study 1, three participants were excluded due to a non-significant linear trend analysis. One participant withdrew his participation when he started the coercion condition as agent (see Supplementary Notes S7 for qualitative observations). On the remaining 86 participants, 28 belonged to the group of junior cadets, 30 belonged to the group of seniors and 27 belonged to the group of privates. The mean age was 23.06 (SD = 3.53). The group of seniors and the group of privates did not differ in the number of years spent in a military organization ($p > 0.2$), which was on average 5.008 years (SD = 1.25). All participants provided written informed consent prior to the experiment. The study was approved by the local ethical committee of the Université libre de Bruxelles (008/2016).

*Material and procedure*: The procedure was almost entirely similar to Study 1. Junior cadets and seniors were recruited and tested at the Royal Military Academy of Belgium (RMA) and privates were recruited and tested either in their usual garrison or at RMA when they happened to be there. In this study, we used a 32-electrodes cap instead of a 64-electrodes cap such as in Study 1 since it was not necessary to have 64 electrodes.

**Reporting summary**. Further information on research design is available in the Nature Research Reporting Summary linked to this article.

## Data availability

Data are available on Open Science Framework: [https://osf.io/8u6hp/]. A reporting summary for this Article is available as a Supplementary Information file. Source data are provided with this paper. Source data are provided with this paper.

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

## Acknowledgements

E.A.C. was supported by the FRS-F.N.R.S (Belgium) and by the Marie Sklodowska-Curie Grant Agreement No. 743685. P.H. was supported by ERC Advanced Grant HUMVOL (323943), and by a Project Grant from Leverhulme Trust (RPG-2016-378). A.C. is a Research Director with the F.R.S.-FNRS (Belgium) and a Senior Fellow of the Canadian Institute for Advanced Research's Brain, Mind and Consciousness program. This work was supported by ERC Advanced Grant RADICAL to A.C. E.A.C., A.C., and P.H. gratefully acknowledge additional support from Evens Foundation in the form of the Evens Science Prize.

## Author contributions

E.A.C. developed the study concept. E.A.C., S.L.B., P.A.M.S.d.G, P.H., and A.C. contributed to the study design. Testing and data collection were performed by E.A.C., S.L.B.,

and P.A.M.S.d.G. E.A.C. performed the data analysis and interpretation under the supervision of S.L.B., P.A.M.S.d.G, P.H., and A.C. E.A.C. drafted the manuscript, and S.L.B., P.A.M.S.d.G, P.H., and A.C. provided critical revisions. All authors approved the final version of the manuscript for submission.

## Competing interests

The authors declare no competing interests.
