## [Peer Review File · Nature Communications]

Reviewers' comments:

Reviewer #1 (Remarks to the Author):

Review of manuscript "The effect of military training on the sense of agency and outcome processing under coercion" by Caspar et al.

This is an impressive paper and behavioral approach on a highly relevant topic. The authors extended their previous important work and methodology on the sense of agency under coercion to a new topic that is difficult to study: agency in different conditions of hierarchical organization. In particular military conditions/situations and how military training may alter the sense of agency and its dependence on coercion. For this the authors carried out two studies in carefully selected large groups of people differing in their military training and profession and compared them to non-military controls. They used intentional binding, embedded within the classical Milgram paradigm. Additional data were recorded using EEG.

The authors find that coercion and differences in military experience modulate intentional binding and that military training can correct/alter such effects. It is argued that their data provide insights on a dynamical process underlying sense of agency by showing how specific military training effects modulate agency.

Below I list several questions and concerns I have about the results and discussion and the manuscript more generally that the authors should take into account when preparing a revised version of the manuscript.

Subjective sense of agency and intentional binding: Following their work on intentional binding and coercion in civilian volunteers (reference 13), the authors here use the elegant measure of intentional binding to gauge the sense of agency and extend to responsibility and the sense of agency in the military context. This is, of course, a valid and interesting approach. However, I am concerned how a behavioural measurement, based on interval estimates, is sufficient to estimate the subjective sense of agency. I agree that, as the authors mention, that explicit reports of the sense of agency may be problematic, but additional data using explicit measures on the subjective sense of agency need to be added to the intentional binding data, on a trial by trial basis. Such additional would confirm/extend the existing intentional binding data and allow authors to directly relate their data to aspects involving the subjective aspects of agency, especially since these latter aspects figure prominently the manuscript text (in introduction and discussion). Such additional data should ideally be measured on a trial by trial basis and not at the end of the experiment. Did the explicit data that the authors recorded (i.e. p6) correlate with the intentional binding data?

EEG data:

Whereas the experimental paradigm, the recruited groups, and the findings on intentional binding are solid (but see my previous comment), it is not clear what the EEG study and the recorded data add to the overall story. What was the main scientific reason to add this analysis?

In addition, the authors recorded 64 channel EEG, but only minimally analyzed their data and focus only on analysis of the N1 auditory component. In my opinion, the quality of the paper and reported data would be improved by deleting the EEG data. For example, it was not clear to me what Fig.3b and 5b (topo plots) are adding to the manuscript.

There is much previous work on the N1 component and more complete approaches exist to fully

analyze multichannel EEG data. If the authors decide to leave the EEG data in the manuscript, more extensive EEG analysis about the involved brain processes should be added.

Finally, in both studies, the EEG data do not confirm the behavioural intentional binding findings and even seem to contradict them? This holds especially for the results of study 2.

The experimental manipulation of intentional binding and its integration into a complex behavioural paradigm (Milgram paradigm) was carried out across very relevant and carefully selected conditions/groups. This is an important achievement and based on excellent previous work from this team of authors. However, the authors only briefly refer to Milgram's original paper in the introduction. Have other groups confirmed these results? Have other related paradigms been developed and tested?

Results study 1: Authors used z-transformed data, which is fine. Yet, they should also report original data (for example as supplemental data). This would also be important in order to be able to compare the magnitude of intentional binding in the present data with previously reported data.

The normalized data plotted in Fig.2 for the junior cadets are 0. Does this suggest that people in this group have on average no agency or intentional binding in this group (free choice condition)?

For the data on intentional binding from study 2, do the number of years of military training that have been received correlate with the differential findings and the magnitude of intentional binding in junior and/versus senior cadets? Are such correlations different from those in privates?

The general discussion is quite long and should be shortened (moving less scientific parts to the supplemental). Yet, I was missing a section on the limitations of the study. The authors should be more careful when discussing subjective sense of agency versus intentional binding.

Reviewer #2 (Remarks to the Author):

Comments to the authors

The present manuscript reports results from two studies investigating the effects of social coercion, i.e. following orders, as compared to deciding freely on the sense of agency (SoA) as well as electrophysiological measures of sensory outcome processing. Further, the authors test whether individuals working in contexts marked by different levels of social hierarchical structure (i.e. civilians and differently ranked individuals in the military), are impacted differently by social coercion. In both studies, participants (civilian students and junior cadets in study 1; privates, junior cadets, and senior cadets in study 2) took turns in a task in which they can increase their own monetary benefit by applying painful electric shocks to the other person in the dyad, and hear a tone after choosing to apply electric shocks (or not) to the other individual. In one condition, subjects were free to choose whether or not to apply electric shocks (free choice condition) and in the other, they are ordered to do so by a third individual (coercion condition).

The main outcome measures in both studies are 1) an implicit measure of SoA, i.e. subjective estimates of the temporal interval between choice and tone and 2) the auditory N1, which is associated with auditory outcome processing. Generally, reductions of the estimated temporal interval

for actions perceived to be one's own actions (or in the present case: free choices) are termed intentional binding (IB). The N1 has been shown to be reduced under conditions of coercion in a previous study from the same group.

The central claim that the authors make is that strict hierarchical social contexts such as the military can impede the differentiation of free and non-free choices (reduced coercion effect on IB) for those subjects whose environment is not marked by high responsibility or accountability (i.e. privates and junior cadets in the present case). Furthermore, the authors argue for an impact of military training on electrophysiological measures of outcome processing. Precisely, they state that senior cadets, i.e. individuals who are trained to give orders and be held accountable for their actions present generally increased markers of outcome processing (higher N1 amplitudes), while individuals who are used to following orders show generally reduced N1 amplitudes. Thus, hierarchical social environments are claimed to impact both SoA as well as neural outcome processing.

The present work is embedded into a theoretical, historical, and philosophical context that is highly interesting and aims at fundamental questions of social determinants of responsibility, human agency, and morality. I believe that the description of methods is detailed enough in order for other researchers to reproduce the studies reported. However, while I find the ideas and questions highlighted by this manuscript interesting, I am not convinced that all of the conclusions drawn are valid. I detail my concerns in the Major points below. Furthermore, some aspects of the manuscript were not clear to me, some of which might be due to wording, others due to unclear or missing information (Minor points below).

Major points:

A central claim the authors make is that upon entering the military, the strict hierarchical organization and need to follow orders leads junior cadets to show a reduction of SoA, measurable as a diminished coercion effect on IB. The authors argue that the nevertheless observable N1 reduction under coercion in study 1 (as in civilian students) might be due to the short period of time junior cadets have so far spent in the military and imply that structural brain changes underlying such an effect could require a longer time spent in this environment. As I take it, this is one reason for comparing junior cadets to senior cadets in study 2, as the latter will have spent a longer time in the military than the former, and hence, alterations in the N1 component might be observed after this longer time of training. However, I am not convinced that the observed effects are necessarily attributable to years spent in such environments, as claimed by the authors. The authors themselves state (in the reporting summary) that senior cadets are "even more limited in number than junior cadets", implying a selection process taking place between being a junior and a senior cadet. Thus, it is possible that the differences observed between these groups are due to other factors (personality, etc.) that are already evident in those junior cadets who will later become senior cadets, but not in those who will not achieve this status. In study 1, several psychometric instruments were completed by the participants, and results on these are reported in order to test whether the results of study 1 could be due to any differences in these measures. However, it is unclear whether any such differences exist in study 2, and whether the results could be explained by such differences. Could the authors provide this information so that any potential confounds with regard to personality differences can be ruled out?

Related to my previous point, I would like to note that I think the usage of the word "regain" or "regained" in the context of the observed effects in senior cadets is perhaps too far-reaching, since it implies that those subjects in the group of senior cadets have in fact experienced a sense of agency reduction upon entering the military, which they have later regained. However, we do not know whether the effect observed in junior cadets is due to entering the military or whether it was pre-

existing in those individuals choosing to become officers (although the authors show that some relevant psychometric scales do not correlate with the effects of interest in study 1). Additionally, the term "regain" implies that such a reduction in SoA was later reversed, possibly but not necessarily due to the emphasis on accountability during officer training, as argued by the authors. Yet, the conclusion that this process actually takes place is not warranted, since (self-)selection processes might allow certain individuals to eventually become senior cadets (who were more difficult to recruit than junior cadets), while others (perhaps those who showed lower sense of agency and accountability for own actions before starting their officer training) might not be selected for senior cadet. Hence, I think the authors' claims, while tempting, are to be taken with a grain of salt and this should be reflected in the wording. While this might seem like a minor point, it is part of the larger question regarding the validity of some of the conclusions drawn.

The authors argue that strict hierarchical social situations, like being a private in the military, should impact both implicit measures of SoA, i.e. the intentional binding (IB) effect, as well as electrophysiological measures of sensory outcome processing, i.e. the amplitude of the N1 component. While the authors provide separate statistical tests regarding effects of the experimental factors on these measures (i.e. ANOVAs), they do not elaborate on the question of whether these measures should be correlated in any way and do not report correlations between the central outcome measures within experimental groups. I would assume that subjects experiencing stronger differences in SoA between conditions (larger IB effect) might also display more differentiated neural outcome processing in response to outcomes that are due to freely chose actions vs actions due to commands from others. If no correlations between these measures are to be expected, what are the reasons for this? Are there any arguments speaking against the assumption that SoA and IB should be correlated or are there any methodological reasons impeding tests of this hypothesis?

In the general discussion, the authors relate their findings to results from a study showing that disobeying an expert's advice leads to stronger activity in anterior cingulate cortex (ACC) and superior frontal gyrus than disobeying a non-experts advice (SFG, Suen, Brown, Morck, & Silverstone, 2014). They draw on this study to argue for a potential effect of neural attenuation and neural cost on the dynamics of whether and why certain individuals might be more or less prone to defy authority. However, I could not fully follow their reasoning. One cause of this is that it was not clear to me what general psychological (or physiological) process the authors assume to link the N1 effects observed in the present study (which are responses to auditory action outcomes) to the MRI effects observed by Suen and colleagues, that were observed during choosing to follow or not follow the advice of another person (i.e. not necessarily linked to the outcome). Could the authors please 1) try to clarify this point to help readers follow the reasoning more easily and 2) elaborate on their concept of "neural cost" and the precise processes they assume to underlie social (dis)obedience?

The authors note that in study 2, all three groups (i.e. privates, junior cadets and senior cadets) were tested by what the authors call "another senior captain". I find this somewhat confusing, as the term captain is used before, but not introduced, and it is therefore unclear whether "senior captain" is equivalent to "senior cadet", especially due to the word "another" in the methods section of study 2. For readers who are not familiar with the exact terminology of military ranks it therefore seems like in study 2, the group of "senior cadets" was tested by "another" individual with the same rank, which raises the question of whether the experimental condition was comparable to that of the other two groups where an individual of a higher rank was running the experiment and giving orders. Precisely, if in fact "senior cadet" and "senior captain" are equivalent ranks, the group of senior cadets in study 2 would have been coerced by an individual with the same rank, while junior cadets and privates would have been coerced by an individual with a higher rank, which in the logic of the manuscript, could well impact the results. Perhaps, the issue I am raising here boils down to something negligible that can be resolved by a clear differentiation of the ranks "senior captain" and "senior cadet", in

which case I would retract this criticism. Otherwise, I think it would be beneficial to the manuscript if the authors could argue why they think that equivalence of ranks is not relevant to the results of the study and their interpretation (NB: We do see that there is a coercion effect on IB in the group of senior cadets in study 2, which provides evidence that despite a potential equivalence of military ranks of senior cadets and senior captains, coercion still impacts SoA).

Minor points:

As stated in the methods section to study 1, agents were asked to wait 2 seconds before pressing a key. Did this apply to both conditions, i.e. in the coercive condition were agents instructed to wait 2 seconds after being told to apply a shock or not in order to obtain a noise free baseline? Otherwise, auditory processing could influence the baseline in the coercive condition (although I am aware that this would not have an effect on between-group comparisons).

In line 424, what do the authors mean by "similar personnel strategy issues"? Does this formulation mean that different groups of individuals might make different use of the scale of 1-1000 ms for rating temporal intervals?

In lines 425-426 the authors note that the type of experimenter did not influence the results in study 1, and hence "we decided not to include these two factors in the analyses of Study 2". With type of experimenter being one factor, which is the other factor? Or is this simply a mistake in wording?

In line 463, the authors state that they applied Bonferroni correction to correlations. However, no correlations are reported, so it is not clear what this information relates to.

In line 542, the term "evoked-related" should be "event-related" or "evoked", I assume.

Were any measures taken in order to remove artifacts from the EEG data, such as visual inspection and manual removal of epochs or ICA?

The article by Suen, Brown, Morck, and Silverstone (2014) that is cited in the discussion is not included in the bibliography.

In the reporting summary, it is stated that 32 electrodes were used for EEG recordings, which conflicts with the statement in the main manuscript, where it states that 64 electrodes were used.

In the sampling strategy, there is an error regarding the power of study 2. It says that power was increased to .09, which should be .90, I assume.

In line 247, does the word "similar" in fact mean that the order of conditions was the same for both individuals in a dyad?

On p. 15, line 648, the word "neurally" is misspelled ("neutrally").

Best regards,
David S Stolz

Comments from Reviewer#1

Review of manuscript “The effect of military training on the sense of agency and outcome processing under coercion” by Caspar et al. This is an impressive paper and behavioural approach on a highly relevant topic. The authors extended their previous important work and methodology on the sense of agency under coercion to a new topic that is difficult to study: agency in different conditions of hierarchical organization. In particular military conditions/situations and how military training may alter the sense of agency and its dependence on coercion. For this the authors carried out two studies in carefully selected large groups of people differing in their military training and profession and compared them to non-military controls. They used intentional binding, embedded within the classical Milgram paradigm. Additional data were recorded using EEG. The authors find that coercion and differences in military experience modulate intentional binding and that military training can correct/alter such effects. It is argued that their data provide insights on a dynamical process underlying sense of agency by showing how specific military training effects modulate agency.

→ We thank Reviewer 1 for this description of our work and for the general appreciation of our design and method.

Below I list several questions and concerns I have about the results and discussion and the manuscript more generally that the authors should take into account when preparing a revised version of the manuscript.

C1: Subjective sense of agency and intentional binding: Following their work on intentional binding and coercion in civilian volunteers (reference 13), the authors here use the elegant measure of intentional binding to gauge the sense of agency and extend to responsibility and the sense of agency in the military context. This is, of course, a valid and interesting approach. However, I am concerned how a behavioural measurement, based on interval estimates, is sufficient to estimate the subjective sense of agency. I agree that, as the authors mention, that explicit reports of the sense of agency may be problematic, but additional data using explicit measures on the subjective sense of agency need to be added to the intentional binding data, on a trial by trial basis. Such additional would confirm/extend the existing intentional binding data and allow authors to directly relate their data to aspects involving the subjective aspects of agency, especially since these latter aspects figure prominently the manuscript text (in introduction and discussion). Such additional data should ideally be measured on a trial by trial basis and not at the end of the experiment. Did the explicit data that the authors recorded (i.e. p6) correlate with the intentional binding data?

→ We thank R1 for this comment. Explicit measures of sense of agency on a trial basis would indeed be interesting but the problem is that with our design they will neither be reliable, nor informative. To evaluate explicitly sense of agency on a trial basis, one needs a paradigm that either induces a doubt about who caused the action (for instance, both participants press a key so they are not sure about who actually caused the outcome to happen) or an incongruence on some trials (the keypress sometimes gives an incongruent outcome). In our paradigm, there was absolutely no doubt about who caused the action and the outcome was always fully congruent with the keypress, for ethical reasons: If someone does not feel comfortable to give shocks to another individual, she has to be able to select a button that for sure will not cause any harm to the other person. With the approval that we have at the moment from the ethic board of the university, we are not authorized to have incongruent trials. Therefore, asking participants to rate agency explicitly on a trial basis would not make sense since their answer will always be the same: they are always the author of their own action, which is entirely predictable. In addition, these measurements are strongly influenced by social desirability, specifically when the decision involves a real pain delivered to someone else and not only a fictitious scenario. Thus, similarly to explicit responsibility ratings, there

are no theoretical reasons to expect different answers between groups of participants. That being said, it is very infrequent that explicit and implicit measurements correlate. It has been mentioned in the literature that explicit and implicit measures in fact reflect two different components of sense of agency: a pre-reflective component (measured with implicit measures) and a more reflective component (measured with explicit measures). Neuroimaging data additional also showed that they are associated with different brain networks: Previous studies have shown that intentional binding is associated with increased activity in the supplementary motor area (SMA) (Kühn, Brass, & Haggard, 2013), while explicit judgements of agency are negatively associated with activity in angular gyrus (AG) (Beyer, Sidarus, Fleming, & Haggard, 2018; Chambon, Wenke, Fleming, Prinz, & Haggard, 2013; Farrer et al., 2008; Spengler, von Cramon, & Brass, 2009).

Exploring the relationship between implicit and explicit sense of agency is very interesting at the theoretical level, but we do not have a paradigm that allow to answer this question since our design had to be consistent with ethical considerations and that we cannot rule out the social desirability bias. If we would have indeed recorded explicit agency in our design, we would have had almost no variability in explicit ratings, thus making any assumptions about significant or non-significant correlations unreliable. Correlations were not performed since there was almost no variability in responsibility ratings, as expected based on the above-mentioned issues.

We have now added a paragraph about that aspect in the general discussion (page 11, lines 25-35): *“In this study, we did not evaluate the sense of agency with explicit measurements on a trial-basis, thus introducing a limitation in the interpretation of our explicit measures of responsibility. The main reason is that studying the sense of agency with explicit measures require to induce a source of variability on a trial-basis, such as unexpected action outcomes (Moore, Middleton, Haggard, & Fletcher 2012) or uncertainty about who caused the outcome (Obhi & Hall, 2011). For ethical reasons, action outcomes were always predictable in our paradigm: If someone does not feel comfortable to send shocks to another individual, she has to be able to select a button that for sure will not cause any harm to the other person. In addition, there was no doubt about who caused the outcomes. To explore more reliably if explicit sense of agency is also modulated by the military context and by the military rank, other experimental designs should be developed.”.*

C2: EEG data: Whereas the experimental paradigm, the recruited groups, and the findings on intentional binding are solid (but see my previous comment), it is not clear what the EEG study and the recorded data add to the overall story. What was the main scientific reason to add this analysis? In addition, the authors recorded 64 channel EEG, but only minimally analyzed their data and focus only on analysis of the N1 auditory component. In my opinion, the quality of the paper and reported data would be improved by deleting the EEG data. For example, it was not clear to me what Fig.3b and 5b (topo plots) are adding to the manuscript.

→ In our initial experiment, we measured the amplitude of the auditory N1 to obtain data about outcome processing in both experimental conditions (free-choice and coercive). Downregulation of outcome processing could indeed have consequences of people’s behaviours, since it has been shown for instance in many studies that downregulation of outcome processing when these outcomes involve the suffering of other are correlated with how prosocially people act towards others (e.g. Hein et al. (2010); FeldmanHall et al. (2015)). It was thus of interest for us to understand if obeying orders involves a downregulation of outcome processing or not, and specifically here if outcome processing would also be influenced by the military rank. We have now added this information in the introduction section, on page 3 lines 6-8.

→ We indeed used a 64-channels EEG cap in Study 1 (and a 32-electrodes cap in Study 2). The reason is not theoretical but practical: at the moment of the testing of Study 1, we only had access to a 64-electrodes cap and it was easier to plug to 64 electrodes than to let half of them suspended and not clipped. But indeed, in our case a 32-electrodes cap was sufficient.

C3: There is much previous work on the N1 component and more complete approaches exist to fully analyze multichannel EEG data. If the authors decide to leave the EEG data in the manuscript, more

extensive EEG analysis about the involved brain processes should be added.

→ There are indeed numerous other processes that could be investigated, but we did not design our task based on those additional processes since we hadn't *a priori* hypotheses and we decided to focus 'only' on sense of agency and outcome processing over the tones. For instance, we could have also measured the N2 and the LPP, ERPs associated with the perception of pain (Cheng, Chen, Decety, 2014). Unfortunately, in this task we did not specify our participants that they had to focus strongly on the hand of the victim, which is mandatory to have a clear recording of N2 and LPP, which reflect the empathic response towards others' pain. We did not ask to focus on the victim's hand to avoid creating an additional task to participants who already had to focus on time perception, not blinking, not moving etc. This is something that we are integrating now in new paradigms, with a better control to objectively evaluate that agents actually look at the victim's hand. In the same line, we could also have analyses TFR, specifically mu suppression, since several studies have shown that observing painful stimuli delivered on somebody else's hand involved a greater desynchronization of the mu band in comparison with a non-painful sensation (e.g. Yang, Decety et al., 2009). But numerous studies have shown that mu desynchronization also occurs when an action is performed (e.g. Chartrian et al., 1959; Pfurtscheller et al., 2000; Pineda, 2005; Ritter et al., 2009; Hari & Salmelin, 1997; Mary et al., 2015), thus involving a potential overlap between mu desynchronization during movement (i.e. keypress) and outcome processing (i.e. the electrical stimulation). Longer action-outcome intervals could be used, but then, these longer intervals will be detrimental to assess sense of agency with implicit methods (Humphrey & Buehner, 2014). Therefore, since we had no previous precise hypothesis about other brain processes at that moment and that we did not design our task to analyse them, if we include some here, the first problem is that it will not be reliable (since not properly designed for) and can be taken as a fishing expedition since we did not create a priori hypothesis on those additional processes. But we agree that there is still a lot of work to do on this topic with many additional brain processes that could be at play, and we are working on some new ones at the moment.

C4: Finally, in both studies, the EEG data do not confirm the behavioural intentional binding findings and even seem to contradict them? This holds especially for the results of study 2.

→ Thank you for this really good question. In fact, several studies have shown that sense of agency and outcome processing relate somehow, in the sense that they may be influenced by similar factors (e.g. incongruence of outcomes) but those studies never showed evidences for reliable statistical correlations. We could thus not predict that they would be correlated, and this is why we analysed them as two different processes. We have now added this aspect in the introduction as follow to increase clarity (page 3, lines 34-37): *"The literature has shown that sense of agency and outcome processing can be related (e.g. Kuhn, Nenchev, Haggard, Brass et al., 2011; Bednark & Franz, 2014) but strong statistical correlations have barely been reported. A military environment could thus have a different effect on outcome processing and on sense of agency"*.

We have also added more precise predictions in the discussion of study 1 about study 2, as follow (page 6, lines 44-49): *"We also predicted that outcome processing could be influenced by military rank (i.e., officers vs privates) after 5 years of military training. Since the function of privates is mostly to execute orders from the military hierarchy, downregulation of outcome processing could be observed in both the free-choice and the coercive condition. On the other hand, senior officers might preserve an upregulation of outcome processing in the free-choice condition, consistently with their function implying commanding subordinates and being accountable for their actions."*

Of note, in a more recent study one of us has performed correlations between sense of agency and different outcome processing measure, such as the auditory N1 but also empathy-related ERPs. Without spoiling too much the results (that we cannot unfortunately mention here because there are not published yet), the sense of agency appears to correlate with some ERPs (i.e. not auditory ERPs) but not others, suggesting that there is actually a link between sense of agency and outcome processing, but not any kind of outcome processing. This will thus be included more consistently in future studies.

C5: The experimental manipulation of intentional binding and its integration into a complex behavioural paradigm (Milgram paradigm) was carried out across very relevant and carefully selected conditions/groups. This is an important achievement and based on excellent previous work from this team of authors. However, the authors only briefly refer to Milgram's original paper in the introduction. Have other groups confirmed these results? Have other related paradigms been developed and tested?

→ We thank R1 for this comment. We indeed generally try to avoid mentioning too much Milgram because he focused on the experimental factors that would modulate the disobedience rates of participants while we focus on the mechanisms by which coercion influences moral behaviours. Even if more recent variants of the Milgram experiment have been conducted (Slater et al. 2006 for an alternative using virtual reality), to our best knowledge, a military population has never been tested in an obedience-like paradigm, despite its main relevance. A first reason is that previous researchers since Milgram tried to avoid working on such tricky topic because of the ethical issues associated with Milgram's studies. The second reason is probably the difficulty associated with recruiting military staff for experimental studies. Even with a contact person in the Belgian army who strongly helped in the realisation of the present study, testing military members also require approval from their own hierarchy, which can refuse their subordinates to participate (we faced this refusal for two other categories, including the one suggested by the editorial team).

C6: Results study 1: Authors used z-transformed data, which is fine. Yet, they should also report original data (for example as supplemental data). This would also be important in order to be able to compare the magnitude of intentional binding in the present data with previously reported data.

→ That is indeed a relevant comment, but the only issue is that non-z-scored data would not allow a reliable comparison between different groups tested in previous studies, since, such as in this experiment, numbers that are provided for interval estimates depend on personal strategy. We could indeed have reported z-scores in previous studies but since we never compared groups before, we did not consider it as relevant. With z-scores provided here, future studies could directly compare to the present study by also using z scores.

C7: The normalized data plotted in Fig.2 for the junior cadets are 0. Does this suggest that people in this group have on average no agency or intentional binding in this group (free choice condition)?

→ The data should always be interpreted in comparison with another condition or group, and not individually. 0 does not mean that they have 0 agency (since higher z-scores mean even lower sense of agency) but rather that they have a lower agency than the civilian group in the free-choice condition. But we assume that experiencing "0" agency when one is performing an action is very unlikely. Agency can be highly reduced, but in a healthy adult population, a minimal experience of agency should still be present.

C8: For the data on intentional binding from study 2, do the number of years of military training that have been received correlate with the differential findings and the magnitude of intentional binding in junior and/versus senior cadets? Are such correlations different from those in privates?

→ We agree that it would be very interesting to observe a correlation between the number of years and interval estimates. The problem is that we cannot perform correlations with the number of years on these two groups: junior cadets were all in year 1 at the moment of the testing and senior cadets we have 1 participant with 7 years, 3 with 6 and all the other are in the system since 5 years. There is thus no variability that would allow to perform such correlations. The group composed of privates had nonetheless more variability, and we have thus run a correlation between the number of years in the military system and z scores for interval estimates in both the free choice and the coercive condition. For both conditions, the correlation was not significant (all $p_s > .1$).

C9: The general discussion is quite long and should be shortened (moving less scientific parts to the supplemental). Yet, I was missing a section on the limitations of the study. The authors should be more

careful when discussing subjective sense of agency versus intentional binding.

→ We thank R1 for this comment. We have tried to reduce the length of the discussion, for instance by removing the part on disobedience and the work of Sun et al. 2014 since it was not highly clear and not the main point of the present study.

R1 is right in the sense that we should have provided more information about how sense of agency and intentional binding are linked. In order to avoid adding too much in the discussion section, we have now added this information in the introduction (page 2, lines 33-41): *“The relationship between time perception and sense of agency is mediated by the involvement of striatal dopaminergic activity, which is crucial for time perception (e.g. Moore et al., 2010; Meck, 2006) and is also driving information from basal ganglia to frontal motor areas (Cunnington et al., 2001; Nachev, Kennard, & Husain, 2008), key brain regions in generating sense of agency (e.g. Kühn, Brass, & Haggard, 2013; Haggard & Whitford, 2004). In classic studies asking participants to estimate the duration of intervals between an action and its predictable outcome, participants report shorter interval estimates when the action was performed voluntarily than when this action was performed involuntarily, for instance after a TMS pulse over the motor cortex (Haggard et al., 2002).”*

Comments from Reviewer#2

Comments to the authors

The present manuscript reports results from two studies investigating the effects of social coercion, i.e. following orders, as compared to deciding freely on the sense of agency (SoA) as well as electrophysiological measures of sensory outcome processing. Further, the authors test whether individuals working in contexts marked by different levels of social hierarchical structure (i.e. civilians and differently ranked individuals in the military), are impacted differently by social coercion. In both studies, participants (civilian students and junior cadets in study 1; privates, junior cadets, and senior cadets in study 2) took turns in a task in which they can increase their own monetary benefit by applying painful electric shocks to the other person in the dyad, and hear a tone after choosing to apply electric shocks (or not) to the other individual. In one condition, subjects were free to choose whether or not to apply electric shocks (free choice condition) and in the other, they are ordered to do so by a third individual (coercion condition).

The main outcome measures in both studies are 1) an implicit measure of SoA, i.e. subjective estimates of the temporal interval between choice and tone and 2) the auditory N1, which is associated with auditory outcome processing. Generally, reductions of the estimated temporal interval for actions perceived to be one's own actions (or in the present case: free choices) are termed intentional binding (IB). The N1 has been shown to be reduced under conditions of coercion in a previous study from the same group.

The central claim that the authors make is that strict hierarchical social contexts such as the military can impede the differentiation of free and non-free choices (reduced coercion effect on IB) for those subjects whose environment is not marked by high responsibility or accountability (i.e. privates and junior cadets in the present case). Furthermore, the authors argue for an impact of military training on electrophysiological measures of outcome processing. Precisely, they state that senior cadets, i.e. individuals who are trained to give orders and be held accountable for their actions present generally increased markers of outcome processing (higher N1 amplitudes), while individuals who are used to following orders show generally reduced N1 amplitudes. Thus, hierarchical social environments are claimed to impact both SoA as well as neural outcome processing.

The present work is embedded into a theoretical, historical, and philosophical context that is highly interesting and aims at fundamental questions of social determinants of responsibility, human agency, and morality. I believe that the description of methods is detailed enough in order for other researchers to

reproduce the studies reported. However, while I find the ideas and questions highlighted by this manuscript interesting, I am not convinced that all of the conclusions drawn are valid. I detail my concerns in the Major points below. Furthermore, some aspects of the manuscript were not clear to me, some of which might be due to wording, others due to unclear or missing information (Minor points below).

→ We thank R2 for this very complete description of our work and for the general appreciation of the present paper.

Major points:

C10: A central claim the authors make is that upon entering the military, the strict hierarchical organization and need to follow orders leads junior cadets to show a reduction of SoA, measurable as a diminished coercion effect on IB. The authors argue that the nevertheless observable N1 reduction under coercion in study 1 (as in civilian students) might be due to the short period of time junior cadets have so far spent in the military and imply that structural brain changes underlying such an effect could require a longer time spent in this environment. As I take it, this is one reason for comparing junior cadets to senior cadets in study 2, as the latter will have spent a longer time in the military than the former, and hence, alterations in the N1 component might be observed after this longer time of training. However, I am not convinced that the observed effects are necessarily attributable to years spent in such environments, as claimed by the authors. The authors themselves state (in the reporting summary) that senior cadets are “even more limited in number than junior cadets”, implying a selection process taking place between being a junior and a senior cadet. Thus, it is possible that the differences observed between these groups are due to other factors (personality, etc.) that are already evident in those junior cadets who will later become senior cadets, but not in those who will not achieve this status. In study 1, several psychometric instruments were completed by the participants, and results on these are reported in order to test whether the results of study 1 could be due to any differences in these measures. However, it is unclear whether any such differences exist in study 2, and whether the results could be explained by such differences. Could the authors provide this information so that any potential confounds with regard to personality differences can be ruled out? Related to my previous point, I would like to note that I think the usage of the word “regain” or “regained” in the context of the observed effects in senior cadets is perhaps too far-reaching, since it implies that those subjects in the group of senior cadets have in fact experienced a sense of agency reduction upon entering the military, which they have later regained. However, we do not know whether the effect observed in junior cadets is due to entering the military or whether it was pre-existing in those individuals choosing to become officers (although the authors show that some relevant psychometric scales do not correlate with the effects of interest in study 1). Additionally, the term “regain” implies that such a reduction in SoA was later reversed, possibly but not necessarily due to the emphasis on accountability during officer training, as argued by the authors. Yet, the conclusion that this process actually takes place is not warranted, since (self-)selection processes might allow certain individuals to eventually become senior cadets (who were more difficult to recruit than junior cadets), while others (perhaps those who showed lower sense of agency and accountability for own actions before starting their officer training) might not be selected for senior cadet. Hence, I think the authors’ claims, while tempting, are to be taken with a grain of salt and this should be reflected in the wording. While this might seem like a minor point, it is part of the larger question regarding the validity of some of the conclusions drawn.

→ We thank R2 for this comment, which is indeed highly relevant. The editorial team suggested an additional study including soon-to-be-cadets, but it was unfortunately impossible to have access to this sample:

- When “soon-to-be-cadets” apply to the Royal Military Academy, they do not sign documents that would allow experimenters to contact them. With GDPR in Europe, we can’t ask the persons in charge of the inscription to provide us a list with personal information.
- A number of those “soon-to-be-cadets” are actually minors, given the fact that they can apply

before their majority, but cannot start the training under 18 (the Belgian legal majority age). We do not have the permission from the ethic committee to adapt our paradigm to non-adults. Since the paradigm involves shocks and money, this is a tricky aspect for non-adults (and also for their parents, which should provide explicit permission).

- In case of testing these future cadets, we would have difficulties to control their testing environment. As a reminder, in our task we tested each group in its own educational system (military at the Royal Military Academy or in military garrison) and civilians at the University. For future cadets, they are on holidays during the months of recruitment (next ones will be June/July 2020) and are not attached to any educational system yet. They are dispersed everywhere in Belgium and Luxembourg and will only be reunited when they start their training at the Royal Military Academy.
- In addition, we have to test pairs of participants. It thus means that we should make those people come to a neutral place by pair. But given that they come from everywhere in Belgium and Luxembourg, we will never be able to create those pairs at a similar location for all testing.

Those issues are unfortunately serious and all together make this proposed experiment in practice almost impossible. We have also checked for testing these soon-to-be-cadets on the very first days of their training. Unfortunately, they start with a 6-week military camp with a schedule full from 8AM to 10PM. We are afraid that testing this category is not an option.

However, we are very sensitive to the question of the pre-selection bias because Reviewer 2 is right when suggesting that it could influence our results. In our study, the selection bias can occur at two levels:

- When individuals start their military training: does the military environment impact the sense of agency or is it already individuals with a low sense of agency that apply for a military job?
- From junior cadets to senior cadets: considering the high exclusion rate in the military, it could be the case that only those with already a high sense of agency and outcome processing persevere up to the level of officer.

In order to answer those two questions, we have now performed new statistical analysis on our data.

- From junior cadets to senior cadets: We have been authorised to know if the junior cadets that we had tested in our two studies had been in the meantime excluded from the military system or not. We have thus separated our group of junior cadets (N=66 after exclusion) in two distinct groups: those who persevered in the military system after the first year and those who dropped out/were expelled. Basically, if a pre-selection bias account for our results on senior cadets, we should have observed that those who were excluded had a lower sense of agency in the free-choice condition and those who remained had a higher sense of agency. In the manuscript, we have reported the results of these analyses on pages 8-9 in the manuscript and in Supplemental Information S6. Results mainly indicated that those who dropped out/were expelled are actually those who had a higher amplitude of the auditory N1 in both experimental conditions, suggestion that initial downregulating outcome processing is a crucial step to progress within a military system. We also observed that those who stayed in the military system had a lower sense of agency than our group of senior cadets in the free-choice condition. It thus emphasizes that an officer training type has a beneficial impact on both outcome processing and sense of agency. Only those who downregulated outcome processing and sense of agency when they were junior cadets actually persevered to become officers. It is thus very unlikely that a pre-selection bias explains the difference between junior and senior cadets.
- When individuals start their military training: Reviewer 2 already emphasised that despite the fact that we observed differences between junior cadets and civilians for some personality traits, those did not account for the difference in sense of agency between those two groups. However, to try to provide a better evidence, we have re-contacted the civilian participants that took part in the study and we asked them three additional questions, assessing to what extent they have ever consider

joining the army, to what extent they think it would be difficult for them to work in a highly hierarchical system and to what extent they would be suited for military environment. None of the answers significantly influenced the results on the sense of agency, neither in the free-choice nor in the coercive condition (see page 5 §1 and Supplemental Information S3). In short, we did not find convincing evidence that reduced sense of agency in people who choose to undergo military training is related to an intrinsic predispositions or personality traits.

C11: The authors argue that strict hierarchical social situations, like being a private in the military, should impact both implicit measures of SoA, i.e. the intentional binding (IB) effect, as well as electrophysiological measures of sensory outcome processing, i.e. the amplitude of the N1 component. While the authors provide separate statistical tests regarding effects of the experimental factors on these measures (i.e. ANOVAs), they do not elaborate on the question of whether these measures should be correlated in any way and do not report correlations between the central outcome measures within experimental groups. I would assume that subjects experiencing stronger differences in SoA between conditions (larger IB effect) might also display more differentiated neural outcome processing in response to outcomes that are due to freely chose actions vs actions due to commands from others. If no correlations between these measures are to be expected, what are the reasons for this? Are there any arguments speaking against the assumption that SoA and IB should be correlated or are there any methodological reasons impeding tests of this hypothesis?

→ This comment has also been raised by reviewer 1 (see C4) and we have now elaborated more in the manuscript about these notions and their potential relationship in the manuscript. We have now added this aspect in the introduction as follow to increase clarity (page 3, lines 34-37): *“The literature has shown that sense of agency and outcome processing can be related (e.g. Kuhn, Nenchev, Haggard, Brass et al., 2011; Bednark & Franz, 2014) but strong statistical correlations have barely been reported. A military environment could thus have a different effect on outcome processing and on sense of agency”*.

We have also added more precise predictions in the discussion of study 1 about study 2, as follow (page 6, lines 44-49): *“We also predicted that outcome processing could be influenced by military rank (i.e., officers vs privates) after 5 years of military training. Since the function of privates is mostly to execute orders from the military hierarchy, downregulation of outcome processing could be observed in both the free-choice and the coercive condition. On the other hand, senior officers might preserve an upregulation of outcome processing in the free-choice condition, consistently with their function implying commanding subordinates and being accountable for their actions.”*

Since no previous studies clearly reported a link between implicit SoA and auditory N1 amplitude, we had no theoretical reason to analysed them as a whole. However, as also mentioned in C4, it appears that SoA measurements correlate with some processes but not others, which would suggest that SoA do not correlate with any kind of outcome processing. But unfortunately this paper is in a too early review stage to be cited here.

C12: In the general discussion, the authors relate their findings to results from a study showing that disobeying an expert’s advice leads to stronger activity in anterior cingulate cortex (ACC) and superior frontal gyrus than disobeying a non-expert advice (SFG, Suen, Brown, Morck, & Silverstone, 2014). They draw on this study to argue for a potential effect of neural attenuation and neural cost on the dynamics of whether and why certain individuals might be more or less prone to defy authority. However, I could not fully follow their reasoning. One cause of this is that it was not clear to me what general psychological (or physiological) process the authors assume to link the N1 effects observed in the present study (which are responses to auditory action outcomes) to the MRI effects observed by Suen and colleagues, that were observed during choosing to follow or not follow the advice of another person (i.e. not necessarily linked to the outcome). Could the authors please 1) try to clarify this point to help readers follow the reasoning more easily and 2) elaborate on their concept of “neural cost” and the precise processes they assume to underlie social (dis)obedience?

→ We agree that indeed this point was not very clear. In order to fit with reviewer 1's comment to reduce the discussion, we have now removed this part because it was not the most interesting point of discussion based on the research questions raised in the present paper.

C13: The authors note that in study 2, all three groups (i.e. privates, junior cadets and senior cadets) were tested by what the authors call "another senior captain". I find this somewhat confusing, as the term captain is used before, but not introduced, and it is therefore unclear whether "senior captain" is equivalent to "senior cadet", especially due to the word "another" in the methods section of study 2. For readers who are not familiar with the exact terminology of military ranks it therefore seems like in study 2, the group of "senior cadets" was tested by "another" individual with the same rank, which raises the question of whether the experimental condition was comparable to that of the other two groups where an individual of a higher rank was running the experiment and giving orders. Precisely, if in fact "senior cadet" and "senior captain" are equivalent ranks, the group of senior cadets in study 2 would have been coerced by an individual

with the same rank, while junior cadets and privates would have been coerced by an individual with a higher rank, which in the logic of the manuscript, could well impact the results. Perhaps, the issue I am raising here boils down to something negligible that can be resolved by a clear differentiation of the ranks "senior captain" and "senior cadet", in which case I would retract this criticism. Otherwise, I think it would be beneficial to the manuscript if the authors could argue why they think that equivalence of ranks is not relevant to the results of the study and their interpretation (NB: We do see that there is a coercion effect on IB in the group of senior cadets in study 2, which provides evidence that despite a potential equivalence of military ranks of senior cadets and senior captains, coercion still impacts SoA).

→ Sorry for this lack of clarity. By "another senior captain", we meant an officer of the rank of senior captain, but a different one than in study 1. Additionally, we named "senior cadets" those participants who are cadets in their fifth year of training at the Royal Military Academy, to make a distinction with the cadets in their first year (the junior cadets). "Senior captain" is a Belgian military rank between "Captain" and "Major", which does not exist in many countries.

Thus, senior cadets and senior captain are two very different populations. The first are still in training to become officers, whereas the latter are already confirmed officers (with about 10 years of experience after their graduation from the Royal Military Academy). This is now explained better on page 7, lines 5-7.

Minor points:

C14: As stated in the methods section to study 1, agents were asked to wait 2 seconds before pressing a key. Did this apply to both conditions, i.e. in the coercive condition were agents instructed to wait 2 seconds after being told to apply a shock or not in order to obtain a noise free baseline? Otherwise, auditory processing could influence the baseline in the coercive condition (although I am aware that this would not have an effect on between-group comparisons).

→ Sorry again for this lack of information. They were instructed to wait 2 s before pressing a key event after receiving the coercive condition. So normally our baseline, which is taken -500 to -300 before the tone is produced. This information was indeed missing in the manuscript, so we now have added it on page 14, lines 1-5.

C15: In line 424, what do the authors mean by "similar personnel strategy issues"? Does this formulation mean that different groups of individuals might make different use of the scale of 1-1000 ms for rating temporal intervals?

→ Even at the individual level, some individuals may for instance prefer lower numbers while other will preferably use higher number. It does not mean that the former have a stronger agency than the later, it simply reflects personal strategy. This is why we used z-scores.

C16: In lines 425-426 the authors note that the type of experimenter did not influence the results in study 1, and hence “we decided not to include these two factors in the analyses of Study 2”. With type of experimenter being one factor, which is the other factor? Or is this simply a mistake in wording?

→ This is indeed a mistake in the wording. We have now corrected this sentence as follow: “*Given that the type of experimenter did not influence the results in Study 1, we decided not to include this factor in the analyses of Study 2. Participants were only tested by a ranked officer who had a similar rank than the officer who was the experimenter in Study 1 (i.e. senior captain).*”

C17: In line 463, the authors state that they applied Bonferroni correction to correlations. However, no correlations are reported, so it is not clear what this information relates to.

→ We are sorry, it was indeed a mistake when we copied-pasted the structure of the results section from Study 1, we have forgotten to remove this sentence. This has now been corrected.

C18: In line 542, the term "evoked-related" should be "event-related" or "evoked", I assume.

→ Thank you for noticing this mistake. We have now corrected this word.

C19: Were any measures taken in order to remove artifacts from the EEG data, such as visual inspection and manual removal of epochs or ICA?

→ We have now added this information in the manuscript: “*Artefacts due to eye movements were removed based on a visual inspection with the removal of epochs containing eye blinks or ocular saccades.*”

C20: The article by Suen, Brown, Morck, and Silverstone (2014) that is cited in the discussion is not included in the bibliography.

→ Thank you for noticing. We have now re-checked entirely the references.

C21: In the reporting summary, it is stated that 32 electrodes were used for EEG recordings, which conflicts with the statement in the main manuscript, where it states that 64 electrodes were used.

→ In fact, we used 64 electrodes in Study 1 and 32 in Study 2. We have now clearly indicated that in the manuscript: “*In this study, we used a 32-electrodes cap instead of a 64-electrodes cap such as in Study 1 since it was not necessary to have 64 electrodes.*”

C22: In the sampling strategy, there is an error regarding the power of study 2. It says that power was increased to .09, which should be .90, I assume.

→ Indeed, thank you for noticing, this has been corrected.

C23: In line 247, does the word “similar” in fact mean that the order of conditions was the same for both individuals in a dyad?

→ This is correct. To increase clarity, we have now modified this sentence as follow: “*The order of free-choice and coercive conditions was counterbalanced but similar within each dyad, meaning that the order of conditions was the same for the two participants in a dyad.*”

C24: On p. 15, line 648, the word “neurally” is misspelled (“neutrally”).

→ Thank you for noticing.

Best regards,
David S Stolz

Reviewers' comments:

Reviewer #1 (Remarks to the Author):

I reiterate that this is an impressive study and behavioural approach on a highly relevant topic. The authors responded to many of my comments. However, I am still concerned about the evaluation of the sense of agency (explicit and implicit), the N1 erp data (and the variable relation to behavioural measures) and how this presented in the revised manuscript. Despite the unresolved concerns listed below, I encourage the authors to make another effort in revising the manuscript.

C1

I am not convinced by the authors answer and still concerned how interval estimates can be sufficient to estimate the subjective sense of agency, especially given the prominent discussion of the subjective aspects of agency in the manuscript. A measure of the sense of agency by an explicit measure is still missing and I disagree that such a measure would be uninformative.

I was actually not asking the authors to alter their paradigm by adding different action consequences with for example incongruent trials (which may be an entirely new project, requiring very careful ethics). Without changing any action outcome, I wondered whether explicit agency ratings may show a coercion modulation on the subjective sense of agency (that may additionally differ across tested populations)? This evaluation would be important to add. Otherwise the authors should modify the manuscript more strongly and avoid emphasis on subjective and sense of agency.

I had asked whether the authors had performed correlation analysis between explicit judgements (those they had already measured as I had understood) and time estimates? Could not find this in their response.

C2-3-4

The authors added a short helpful statement to the revised introduction. I am concerned though that they responded to my question C3 as if a more detailed analysis of the N1 would amount to a "fishing expedition". Again, the N1 component in ERPs has been linked to many other perceptual, motor and cognitive aspects beyond outcome processing. It may not reflect outcome processing, but any of the other processes. In addition, not many readers may be familiar with the term outcome processing. This should be clearly expressed in the revision. Please clarify these issues in a re-revision.

Has the present study been pre-registered?

Finally, the variable relationship between N1 and intentional binding (C4) remains unclear, and adds to the complexity of factors likely at work in this study.

Reviewer #2 (Remarks to the Author):

Comments to authors

I appreciate the authors' efforts to improve the manuscript and to address the concerns raised by reviewer 1 and myself. Below, I will first comment on the extent to which I believe the authors responses and extensions to the manuscript satisfy the old concerns raised. Second, I shortly note a point regarding the discussion section that I could not entirely follow.

Old concerns:

Regarding the potential limitations due to possible selection biases, I appreciate the authors' efforts in explaining why they were not able to conduct the study suggested by the editorial team. I also appreciate the efforts shown by re-contacting the civilian participants in order to gather more data and for the additional analyses that are now contained in the manuscript with the goal of ruling out – as far as possible – the possibility of selection biases. I do think that these analyses have added to the quality of the manuscript and provide relevant evidence in support of the authors' main claims.

Regarding the concern raised by reviewer 1, i.e. whether there is a need for explicit, trial-by-trial sense of agency (SoA) ratings, I can follow the authors' reasoning that ethical limitations on the study design (subjects need to be able not to shock, if they don't want to) impede the usefulness of such ratings for the present experimental protocol. Yet, a closer understanding of the exact relationship between intentional binding, explicit ratings of sense of agency and outcome processing would indeed be desirable (both on conceptual levels and with regard to biological correlates and mechanisms), but I take it that it is not the focus of the presented experiments to provide deeper insights into this. Thus, I think that the present data suffice for demonstrating that (implicit) sense of agency and outcome processing can be impacted by (extreme forms of) social hierarchies such as the military. Put differently, the data demonstrate that the previously introduced "coercion effect", i.e. the parallel reduction of N1 amplitude and "stretching" of perceived time intervals (intentional binding), are influenced by one's social environment. However, I assume the exact (psychological/conceptual/biological) mechanisms linking (explicit and implicit) measures of sense of agency to neural outcome processing need to be addressed in other studies. I think that the manuscript would profit from a at least a short statement in the discussion that reiterates this point, picking up the one in the introduction (lines 134-137), and again pointing to relevant literature or open research questions. Otherwise, sense of agency and outcome processing appear somewhat side-by-side, with some implied link between them, that however remains unresolved at the end of the draft.

Related to the previous point, I think that the N1 data serve to demonstrate that neural outcome processing may potentially be affected by environments marked by strong hierarchical organization. Yet, I agree with reviewer 1 that it is not clear what the topographic plots add to the story of the manuscript. Perhaps, in order to reduce the impression that the topographies are of central relevance, moving them to the supplements or reducing their size in relation to the N1 bar plots could help streamlining the appearance of the draft and aid pointing the readers in the right direction.

I believe that removing the paragraph on neural cost from the general discussion has improved the clarity of the manuscript.

I would like to thank the authors for addressing my remaining criticisms by clarifying several points that irritated me in the first review round.

New comment:

In the second-to last paragraph of the discussion of study 2 (lines 394-402), the authors, to my understanding, are arguing the possibility that N1 effects are due to differences in education levels between groups. First, they say that differences between privates and seniors could be due to differences in education. Next, they state that despite differences in education between seniors and juniors, there was no difference in the N1 amplitude. Together, I take it, they argue that these disparate findings argue against the possibility that education differences dominantly influenced the results (because then, juniors and privates should be more similar, because they both have considerably lower education than seniors). However, the last sentence of the paragraph appears to

state the opposite (specifically, in lines 401-402): "It is thus more likely that differences between our groups is the mere results of difference in education degree". Shouldn't it rather be: "... it is thus unlikely that the observed differences between these groups are the mere result of differences in education degree"? If the sentence is in fact correct in the way it is currently written in the draft, this would mean that the authors think the N1 effects are due to education, and not to military training.

Best regards,
David S Stolz

Reviewer #1 (Remarks to the Author):

I reiterate that this is an impressive study and behavioural approach on a highly relevant topic. The authors responded to many of my comments. However, I am still concerned about the evaluation of the sense of agency (explicit and implicit), the N1 erp data (and the variable relation to behavioural measures) and how this presented in the revised manuscript. Despite the unresolved concerns listed below, I encourage the authors to make another effort in revising the manuscript.

→ *We would like to thank again R1 for having taken the time to read the resubmitted version of this manuscript and for her/his general appreciation of our study. We have provided below a response to the remaining concerns, which we hope would help to improve again the quality of the manuscript.*

C1

I am not convinced by the authors answer and still concerned how interval estimates can be sufficient to estimate the subjective sense of agency, especially given the prominent discussion of the subjective aspects of agency in the manuscript. A measure of the sense of agency by an explicit measure is still missing and I disagree that such a measure would be uninformative.

I was actually not asking the authors to alter their paradigm by adding different action consequences with for example incongruent trials (which may be an entirely new project, requiring very careful ethics). Without changing any action outcome, I wondered whether explicit agency ratings may show a coercion modulation on the subjective sense of agency (that may additionally differ across tested populations)? This evaluation would be important to add. Otherwise the authors should modify the manuscript more strongly and avoid emphasis on subjective and sense of agency.

→ *The status of interval estimates as a measure of experience of agency has been discussed at length in the literature. The point remains controversial. It is widely known that judgements of agency and responsibility are subject to strong biases, particularly related to social and affective valuation (Bandura, 2006) – this is one reason for preferring implicit measures in many studies. Our argument is that interval estimates are a useful proxy for sense of agency, and that this proxy is less obviously affected by such desirability factors. There are other concerns with explicit agency judgements in situations such as the one we study here. In our task, there is one and only one possible cause of the outcome (shock/no shock) on each trial, namely the key that the participant pressed. In these situations, explicit judgements of agency and responsibility tend to be at ceiling. Explicit judgements of agency may be more informative where there is uncertainty over who or what caused an outcome, or in situations involving series of multiple events over time (e.g., was my pattern of behaviour responsible for my partner's becoming depressed). For these reasons, explicit judgements seemed to us an unreliable way to study the sense of agency for specific individual events occurring under coercion. We have acknowledged that interval estimation is only a proxy for sense of agency and we now clearly state it in the introduction (page 2 line 42) but we maintain it is an important and informative proxy, which allows us to study directly the experience of agency under coercion in a way that has not previously been straightforward.*

I had asked whether the authors had performed correlation analysis between explicit judgements (those they had already measured as I had understood) and time estimates? Could

not finds this in their response.

→ *Sorry for missing this point. We performed correlations across subjects between z scores for interval estimates, explicit ratings of responsibility and the amplitude of the auditory N1. None of those correlations were significant (all ps > .3). Other studies which attempted similar correlations also failed to find a significant relation (Dewey & Knoblich, 2014). However, we emphasise that these are correlations across individual differences between people: they do not address the question (which we think the reviewer has in mind) about whether there is a correlation across EVENTS, or TRIALS, i.e., whether a short perceived interval is associated with an explicit judgement of high agency or responsibility. This correlation would be interesting, but it would be difficult to do because of explicit judgements of agency for events that are not so sensitive (see above). Further, it is not really appropriate for our design. For interest, we have other data, using completely different designs which are appropriate for testing correlation across events, and we do indeed find it – but that is another publication, and still in preparation.*

C2-3-4

The authors added a short helpful statement to the revised introduction. I am concerned though that they responded to my question C3 as if a more detailed analysis of the N1 would amount to a “fishing expedition”. Again, the N1 component in ERPs has been linked to many other perceptual, motor and cognitive aspects beyond outcome processing. It may not reflect outcome processing, but any of the other processes. In addition, not many readers may be familiar with the term outcome processing. This should be clearly expressed in the revision. Please clarify these issues in a re-revision.

We agree that the N1 is indeed sensitive to several factors. In this study, we are of course primarily interested in how the N1 is affected by coercion. The design of our study should allow us to identify only the change in N1 amplitude due to coercion, but we accept the reviewer’s point that we should exclude these other explanations if possible.

- *Regarding the perceptual factor, the tone was always the same, presented with the same intensity to all our participants. Since participants were all young adults aged between 18 and 30 without any physical dysfunctions, we have no reasons to stipulate that the tone was perceived differently between groups. For the sake of clarity, we have now thus emphasized this aspect in the method section, on page 14, lines 17-19: “Whether the agent delivered a shock or not, a 400Hz tone occurred after the key press. The loudness, duration and intensity of the tone was similar for all participants in each experimental condition.”*
- *Regarding the motor aspects, all participants were requested to press the same keyboard with the same two fingers. Thus, motor performance should be similar across participants and thus not explain differences between groups.*
- *Finally, regarding cognitive aspects, we agree that attention might be playing a role, although the literature is quite ambiguous regarding the effect of attention on early ERPs. However, we have no reasons to consider that attention differed between groups in Study 2. First, all participants correctly obeyed orders identically in the coercive condition. A lack of attention would have for instance let to more “incorrect” button press. Second, with the task of interval estimates which requires some focus to be correctly performed, we excluded participants based on a linear trend analysis which would suggest that they did not perform the task correctly. We had to remove only three participants out of 90, from different groups, suggesting that there was no difference in attention between our groups.*

We have now added this information in the discussion section on page 10 from line 17: “Our experiment was designed to identify changes in N1 amplitude due to coercion, and the way

that different groups respond to coercion. However, other studies showed that the amplitude of the auditory N1 could be modulated by other factors. A previous study showed that age, education and intelligence could modulate the amplitude of the auditory N1 [39]. For instance, a higher amplitude of the auditory N1 was observed for high number of years of education. Seniors and privates differ in educational levels: seniors have a university master degree while the privates require only an elementary school certificate, thus potentially explaining differences in outcome processing between seniors and privates. However, the amplitude of the auditory N1 in the free-choice condition did not differ between junior cadets and senior, despite 5 more years of military training. Thus, since duration of military training did not influence N1 amplitudes, it seems unlikely that differences between other groups merely reflect differences in duration of education prior to military service. Other studies showed that the amplitude of the auditory N1 could be modulated by perceptual, motor and cognitive factors [40]. However, it is also unlikely that those factors influence group by coercion interactions in our study. All participants were in the same age range [41], without any physical disabilities, and heard a tone similar in both frequency and loudness across the two experimental conditions [42]. Also, participants all used the same keyboard and the same fingers to press the buttons, ruling out the influence of motor performance on the amplitude of the auditory N1. Attention has been previously discussed as modifying both late and early (i.e. the auditory N1) ERPs. However, it is unlikely that a difference in attention explains the difference between groups since the majority performed correctly the task of interval estimates and did not commit mistakes when pressing the buttons in the coercive condition. Therefore, it seems unlikely that the N1 results in our experimental design are confounded by these other factors, although it cannot be entirely excluded. “

We have also now modified a sentence in the introduction in order to define more clearly the term 'outcome processing' (page 3, lines 4-6): “This suggested that coercive instructions reduce the sensory processing for action outcomes (i.e. outcome processing), thus probably explained why obeying orders can influence social behaviors [e.g. 20,21].”

Has the present study been pre-registered?

→ The study was not pre-registered. We started this study 4 years ago and have to admit that we did not consider this option at that moment.

Finally, the variable relationship between N1 and intentional binding (C4) remains unclear, and adds to the complexity of factors likely at work in this study.

→ No previous studies reported reliable correlations between the binding and the N1. We have now modified the last sentences of the introduction to be clearer: “Previous studies suggest that sense of agency and outcome processing are related because similar factors influence them [e.g. 24,25] but reliable statistical correlations between those measures have barely been reported [26]. Similarly, implicit and explicit measures of agency represent different levels of representations of the self-in-action [26], with distinctive neural bases [27-32]. A military environment could thus have a different effect on outcome processing and on the sense of agency.”

In accordance with R2's comment, we have added another sentence related to this in the discussion section but we have tried not to emphasize too much on this aspect to avoid confusing the readers about the real aim of our paper.

page 12, lines 33-37: “Taken together, our results also highlight a dissociation between the implicit sense of agency and outcome processing. The psychological, conceptual and

biological mechanisms linking both explicit and implicit measures of the sense of agency to neural outcome processing still need to be clarified and addressed in future studies.
“

Reviewer #2 (Remarks to the Author):

Comments to authors

I appreciate the authors' efforts to improve the manuscript and to address the concerns raised by reviewer 1 and myself. Below, I will first comment on the extent to which I believe the authors responses and extensions to the manuscript satisfy the old concerns raised. Second, I shortly note a point regarding the discussion section that I could not entirely follow.

→ *We would like to thank R2 for his time in reading the resubmitted version of our manuscript and for his help in helping us to improve the quality of the publication from the first round of review.*

Regarding the concern raised by reviewer 1, i.e. whether there is a need for explicit, trial-by-trial sense of agency (SoA) ratings, I can follow the authors' reasoning that ethical limitations on the study design (subjects need to be able not to shock, if they don't want to) impede the usefulness of such ratings for the present experimental protocol. Yet, a closer understanding of the exact relationship between intentional binding, explicit ratings of sense of agency and outcome processing would indeed be desirable (both on conceptual levels and with regard to biological correlates and mechanisms), but I take it that it is not the focus of the presented experiments to provide deeper insights into this. Thus, I think that the present data suffice for demonstrating that (implicit) sense of agency and outcome processing can be impacted by (extreme forms of) social hierarchies such as the military. Put differently, the data demonstrate that the previously introduced “coercion effect”, i.e. the parallel reduction of N1 amplitude and “stretching” of perceived time intervals (intentional binding), are influenced by one's social environment. However, I assume the exact (psychological/conceptual/biological) mechanisms linking (explicit and implicit) measures of sense of agency to neural outcome processing need to be addressed in other studies. I think that the manuscript would profit from a at least a short statement in the discussion that reiterates this point, picking up the one in the introduction (lines 134-137), and again pointing to relevant literature or open research questions. Otherwise, sense of agency and outcome processing appear somewhat side-by-side, with some implied link between them, that however remains unresolved at the end of the draft.

→ *We thank R2 for this suggestion and, in accordance with R1's remaining concerns, we have now added a discussion point about that aspect in the general discussion section (page page 12, lines 33-37): “Taken together, our results also highlight a dissociation between the implicit sense of agency and outcome processing. The psychological, conceptual and biological mechanisms linking both explicit and implicit measures of the sense of agency to neural outcome processing still need to be clarified and addressed in future studies.*

We have also modified the last sentences of the introduction to be clearer: “Previous studies suggest that sense of agency and outcome processing are related because similar factors influence them [e.g. 24,25] but reliable statistical correlations between those measures have barely been reported [26]. Similarly, implicit and explicit measures of agency represent different levels of representations of the self-in-action [26], with distinctive neural bases [27-

32]. A military environment could thus have a different effect on outcome processing and on the sense of agency.

Related to the previous point, I think that the N1 data serve to demonstrate that neural outcome processing may potentially be affected by environments marked by strong hierarchical organization. Yet, I agree with reviewer 1 that it is not clear what the topographic plots add to the story of the manuscript. Perhaps, in order to reduce the impression that the topographies are of central relevance, moving them to the supplements or reducing their size in relation to the N1 bar plots could help streamlining the appearance of the draft and aid pointing the readers in the right direction.

→ *For the sake of clarity, we have now removed the topographic plots from the N1 graphs.*

New comment:

In the second-to last paragraph of the discussion of study 2 (lines 394-402), the authors, to my understanding, are arguing the possibility that N1 effects are due to differences in education levels between groups. First, they say that differences between privates and seniors could be due to differences in education. Next, they state that despite differences in education between seniors and juniors, there was no difference in the N1 amplitude. Together, I take it, they argue that these disparate findings argue against the possibility that education differences dominantly influenced the results (because then, juniors and privates should be more similar, because they both have considerably lower education than seniors). However, the last sentence of the paragraph appears to state the opposite (specifically, in lines 401-402): “It is thus more likely that differences between our groups is the mere results of difference in education degree”. Shouldn’t it rather be: “... it is thus unlikely that the observed differences between these groups are the mere result of differences in education degree”? If the sentence is in fact correct in the way it is currently written in the draft, this would mean that the authors think the N1 effects are due to education, and not to military training.

→ *The interpretation of R2 regarding this discussion point is correct and we have now indeed realized that the mentioned sentence was wrongly written and stated the opposite of our point. We thank R2 for this very careful reading of our manuscript and we have now modified this sentence as suggested: “It is thus more unlikely that differences between our groups is the mere results of difference in education degree”.*

Best regards,
David S Stolz

****REVIEWERS' COMMENTS:**

Reviewer #1 (Remarks to the Author):

I thank the authors for their efforts in revising the ms of this impressive study. The data have become even more relevant considering that the entire world seems to be currently living under CoV-2 related coercion. The revised manuscript is much improved, although it is unfortunate that there are no explicit agency judgements. The discussion of the other points is also improved. Happy to accept the paper in its current form.

Reviewer #2 (Remarks to the Author):

Comments to the authors - NCOMMS-19-19198B

I appreciate the authors' efforts to revise and resubmit their manuscript a second time. In sum, it is now clearer which questions are within the scope of the manuscript, and which are not or still unclear in the literature on sense of agency and outcome processing. I will detail my take on a few more specific points below.

Centrally, the paper finds that (extreme forms of) social hierarchies can potentially alter both outcome processing and sense of agency under coercion. In principle, the assignment of study participants to the different groups does not satisfy the requirements of an experimental protocol, since – for obvious reasons – a random group assignment was not possible. Yet, the authors have performed several additional analyses and thus, as far as the study design permits, it may be assumed that the reported effects are in fact due to military training, and not self-selection or other processes.

It is now also clearer that the reported findings on intentional binding and the N1 reflect two different (although potentially somehow related) processes, namely implicit sense of agency (SoA) and outcome processing. Related to concerns raised by the other reviewer and to methodological reasons of the present study, it is not yet clear if and how implicit SoA is linked to explicit responsibility ratings on a trial-by-trial basis. Neither does the article provide correlational evidence linking these processes to outcome processing. This link, however, is not the focus of the manuscript and I felt that this was less clear before. I appreciate the additional clarifications and analyses now provided by the authors that in my opinion help to understand which questions are, and which aren't, focused by the presented studies.

Last, and in relation to a point raised by the other reviewer, the authors now discuss in more detail that the N1 component is known to be susceptible to other factors such as motor functions, cognitive processes such as attention, or education, to name a few, and argue why they believe that these factors are unlikely to have caused the observed effects. In my opinion, the manuscript has been improved by providing some references to the EEG literature and discussions of potential confounds that were missing in earlier drafts. It has thus become more transparent which potential confounds could have affected the EEG findings, and the authors' arguments against the possible influence of such effects is now explicit, and in my view sound.

With respect to the topographical EEG plots, I was somewhat surprised that they are now completely deleted (and as I take it will not be presented in the supplements either?). I assume that this decision was based on the other reviewer's position regarding these plots in addition to my suggestion to reduce their size or move them to the supplements. I would in fact encourage the authors to not

delete these plots entirely, but provide them in the supplementary information, just not as prominently as they appeared before within the main manuscript. In case my previous statement in this regard was misleading, I would like to apologize for being unclear.

The questions targeted in the are very important with regard to ethical and juridical concepts of responsibility for own actions, fundamental philosophical concepts of selfhood, and also in front of the historical and political background of the Nürnberg trials that the authors allude to. It is thus interesting to see that both implicit sense of agency and outcome processing under coercion seem to be susceptible to military training, although their exact interdependence remains to be clarified by future studies.

Best regards,
David S Stolz

****REVIEWERS' COMMENTS:**

Reviewer #1 (Remarks to the Author):

I thank the authors for their efforts in revising the ms of this impressive study. The data have become even more relevant considering that the entire world seems to be currently living under CoV-2 related coercion. The revised manuscript is much improved, although it is unfortunate that there are no explicit agency judgements. The discussion of the other points is also improved. Happy to accept the paper in its current form.

→ We thank Reviewer 1 for her/his time and effort in reviewing this manuscript. We agree that with the current sanitary situation, the data are indeed even more relevant, specifically in a context of individual responsibility.

Reviewer #2 (Remarks to the Author):

Comments to the authors - NCOMMS-19-19198B

I appreciate the authors' efforts to revise and resubmit their manuscript a second time. In sum, it is now clearer which questions are within the scope of the manuscript, and which are not or still unclear in the literature on sense of agency and outcome processing. I will detail my take on a few more specific points below.

Centrally, the paper finds that (extreme forms of) social hierarchies can potentially alter both outcome processing and sense of agency under coercion. In principle, the assignment of study participants to the different groups does not satisfy the requirements of an experimental protocol, since – for obvious reasons – a random group assignment was not possible. Yet, the authors have performed several additional analyses and thus, as far as the study design permits, it may be assumed that the reported effects are in fact due to military training, and not self-selection or other processes.

It is now also clearer that the reported findings on intentional binding and the N1 reflect two different (although potentially somehow related) processes, namely implicit sense of agency (SoA) and outcome processing. Related to concerns raised by the other reviewer and to methodological reasons of the present study, it is not yet clear if and how implicit SoA is linked to explicit responsibility ratings on a trial-by-trial basis. Neither does the article provide correlational evidence linking these processes to outcome processing. This link, however, is not the focus of the manuscript and I felt that this was less clear before. I appreciate the additional clarifications and analyses now provided by the authors that in my opinion help to understand which questions are, and which aren't, focused by the presented studies.

Last, and in relation to a point raised by the other reviewer, the authors now discuss in more detail that the N1 component is known to be susceptible to other factors such as motor functions, cognitive processes such as attention, or education, to name a few, and argue why they believe that these factors are unlikely to have caused the observed effects. In my opinion, the manuscript has been improved by providing some references to the EEG literature and discussions of potential confounds that were missing in earlier drafts.

It has thus become more transparent which potential confounds could have affected the EEG findings, and the authors' arguments against the possible influence of such effects is now explicit, and in my view sound.

→ We thank Reviewer 2 for these very detailed comments.

With respect to the topographical EEG plots, I was somewhat surprised that they are now completely deleted (and as I take it will not be presented in the supplements either?). I assume that this decision was based on the other reviewer's position regarding these plots in addition to my suggestion to reduce their size or move them to the supplements. I would in fact encourage the authors to not delete these plots entirely, but provide them in the supplementary information, just not as prominently as they appeared before within the main manuscript. In case my previous statement in this regard was misleading, I would like to apologize for being unclear.

→ We have now re-introduced them in the figure, in a small size, as suggested.

The questions targeted in the are very important with regard to ethical and juridical concepts of responsibility for own actions, fundamental philosophical concepts of selfhood, and also in front of the historical and political background of the Nürnberg trials that the authors allude to. It is thus interesting to see that both implicit sense of agency and outcome processing under coercion seem to be susceptible to military training, although their exact interdependence remains to be clarified by future studies.

Best regards,
David S Stolz